# Learning-related contraction of gray matter in rodent sensorimotor cortex is associated with adaptive myelination

**Tomas Mediavilla[1], Özgün Özalay[1], Héctor M Estévez-Silva[1], Bárbara Frias[1], Greger Orädd[1], Fahad R Sultan[1], Claudio Brozzoli[2,3], Benjamín Garzón[3,4], Martin Lövdén[3,4], Daniel J Marcellino[1]\***

[1]Department of Integrative Medical Biology, Umeå University, Umeå, Sweden; [2]IMPACT, Centre de Recherche en Neurosciences de Lyon, Lyon, France; [3]Aging Research Center, Department of Neurobiology, Care Sciences and Society, Karolinska Institute, Solna, Sweden; [4]Department of Psychology, University of Gothenburg, Gothenburg, Sweden

**\*For correspondence:**
daniel.marcellino@umu.se;
daniel.marcellino@me.com

**Competing interest:** The authors declare that no competing interests exist.

**Abstract** From observations in rodents, it has been suggested that the cellular basis of learning-dependent changes, detected using structural MRI, may be increased dendritic spine density, alterations in astrocyte volume, and adaptations within intracortical myelin. Myelin plasticity is crucial for neurological function, and active myelination is required for learning and memory. However, the dynamics of myelin plasticity and how it relates to morphometric-based measurements of structural plasticity remains unknown. We used a motor skill learning paradigm in male mice to evaluate experience-dependent brain plasticity by voxel-based morphometry (VBM) in longitudinal MRI, combined with a cross-sectional immunohistochemical investigation. Whole-brain VBM revealed nonlinear decreases in gray matter volume (GMV) juxtaposed to nonlinear increases in white matter volume (WMV) within GM that were best modeled by an asymptotic time course. Using an atlas-based cortical mask, we found nonlinear changes with learning in primary and secondary motor areas and in somatosensory cortex. Analysis of cross-sectional myelin immunoreactivity in forelimb somatosensory cortex confirmed an increase in myelin immunoreactivity followed by a return towards baseline levels. Further investigations using quantitative confocal microscopy confirmed these changes specifically to the length density of myelinated axons. The absence of significant histological changes in cortical thickness suggests that nonlinear morphometric changes are likely due to changes in intracortical myelin for which morphometric WMV in somatosensory cortex significantly correlated with myelin immunoreactivity. Together, these observations indicate a nonlinear increase of intracortical myelin during learning and support the hypothesis that myelin is a component of structural changes observed by VBM during learning.

## Editor's evaluation

This study is a convincing and useful addition to the literature on the role of adaptive myelination during fine-motor learning, addressing the important question of whether learning is associated with cortical structural changes as assessed by longitudinal magnetic resonance imaging (MRI) measurements in mice. Novel findings include the observation that MRI measures of myelination increase rapidly in the early stages of motor learning and then decrease during later stages, consistent with models of learning in which initial rapid neural circuit modification is followed by stabilization and pruning. The authors show that a more direct measure of myelination – myelin basic protein immunoreactivity – also follows this non-linear type of time course. In demonstrating these

learning-related changes, the study also increases confidence that MRI-based metrics can be used to follow myelin changes non-invasively in "real-time".

## Introduction

Longitudinal structural MRI (sMRI) of the human brain has revealed experience-dependent local changes in estimates of gray matter volume (GMV) (*Draganski et al., 2004*). Recent evidence suggests that these changes in GMV follow a nonlinear pattern throughout training (*Wenger et al., 2017b*). The specific biological components that elicit these volumetric changes are not understood. From observations in rodents, it has been suggested that the cellular basis of changes in GMV detected using sMRI may, in part, be due to learning-dependent changes in dendritic spine density (*Keifer et al., 2015*), alterations in astrocyte volume (*Kleim et al., 2007*; *Woo et al., 2018*), and adaptations within intracortical myelin (*McKenzie et al., 2014*; *Keiner et al., 2017*; *Kougioumtzidou et al., 2017*; *Zatorre et al., 2012*). Macroscopic plasticity of brain structure determined by sMRI has been directly associated with neuronal dendritic spine plasticity together with astrocyte reorganization in the absence of cell proliferation (*Keifer et al., 2015*; *Schmidt et al., 2021*), in which these plastic changes were observed to be transient. Furthermore, observations using two-session sMRI (before vs. after training) indicated macrostructural volumetric increases within areas involved in motor control, sensory processing, learning, and memory (*Badea et al., 2019*). At the microstructural level, using ex vivo diffusion tensor imaging (DTI), significant white matter (WM) differences have been observed in WM underlying motor cortex (*Sampaio-Baptista et al., 2013*), as well as experience-dependent DTI differences observed in GM structure (*Sampaio-Baptista et al., 2020*). Estimates of changes in GMV using sMRI reflect a composite mixture of these biological changes. For example, changes in astrocytes or synaptic remodeling may explain changes in regional GMV. However, changes in intracortical myelin may also affect estimates of GMV, such that increases in intracortical myelin may reduce estimates of GMV, and the inverse, for which decreases in intracortical myelin may increase estimates of GMV. Despite more recent developments of quantitative MRI (*Natu et al., 2019*; *Weiskopf et al., 2013*), the mix and interplay between GM and WM changes, with respect to time from the start of the learning paradigm, that result in nonlinear, experience-dependent, adaptive brain responses observed both in rodents and human studies remain unknown.

Myelin plasticity is critical for learning and memory, and myelin plasticity driven by both neuronal activity and experience has been described (*Gibson et al., 2014*; *Mensch et al., 2015*; *Swire and Ffrench-Constant, 2018*; *Bacmeister et al., 2020*; *Wake et al., 2011*; *Makinodan et al., 2012*). Active myelination by newly recruited oligodendrocytes has been shown to be necessary for learning and memory (*McKenzie et al., 2014*; *Geraghty et al., 2019*; *Pan et al., 2020*; *Steadman et al., 2020*). In addition, the learning of a novel skilled reaching task in rodents is associated with functional reorganization of cortical motor maps, including an expanded representation of the trained limb (*Kleim et al., 1998*; *Kleim et al., 2004*; *Tennant et al., 2011*). This functional remapping is accompanied by a variety of structural and functional changes, including synaptogenesis, increased spine formation, and glial changes (*Kleim et al., 2004*; *Xu et al., 2009*). It was very recently described that learning in a forelimb skilled-reaching paradigm transiently suppresses oligodendrogenesis while increasing oligodendrocyte precursor cell (OPC) differentiation, oligodendrocyte maturation, and myelin sheath remodeling in forelimb motor cortex (*Bacmeister et al., 2020*). Learning-induced suppression of oligodendrocytes was transient but left OPC differentiation unaffected, which suggests that learning may temporarily decrease survival and integration of differentiated OPCs as mature myelinating oligodendrocytes (*Bacmeister et al., 2020*), in line with previous work in the developing central nervous system (*Hill et al., 2014*). However, it is unknown whether adaptive myelination is restricted to discrete brain areas to enable fine-tuning of adaptive circuit responses. Furthermore, the temporal dynamics of myelin plasticity and how it relates to sMRI-based measurements of structural plasticity remain unknown.

In this study, we used the single-pellet skilled reaching task, a well-established paradigm for motor skill learning research in rodents (*Whishaw et al., 1986*), combined with longitudinal MRI and cross-sectional immunohistochemical analyses, to evaluate the time course of experience-dependent brain changes and the related links between micro- and macrostructural changes. To analyze MRI data, we used voxel-based morphometry (VBM), a key technique to evaluate macroscopic changes in GMV

that is frequently used to investigate a broad spectrum of neurological processes spanning from learning (*Badea et al., 2019*) and memory (*Keifer et al., 2015*) to neurodegeneration (*Kim et al., 2020*; *Sawiak et al., 2009*) and cognitive impairment (*Bagdatlioglu et al., 2020*). This morphometric technique calculates voxel-wise estimates of GMV, and in the cortex, GMV is dependent upon local cortical thickness and surface area (*Ashburner and Friston, 2000*), and on local tissue composition (*Asan et al., 2021*). Studies in brain plasticity using sMRI often rely on T1-weighted MR imaging that are sensitive to myelin, rendering the signal of a single voxel highly dependent on the presence of myelin, thereby influencing the estimated volume.

Gray matter (GM) and WM are classical anatomical terms to denote myelin-poor and myelin-rich regions of the brain. As part of the VBM analysis, the brain is segmented into different components reflecting GM, WM, and cerebral spinal fluid (CSF) with the use of study-specific tissue priors. The metric used for comparison in VBM depends in part on the water content in a voxel, which is influenced by the fraction of myelin to other tissue. GM regions of the brain contain myelin to a varying degree (e.g., intracortical myelin) and thus, both GMV and WMV can be estimated in GM regions of the brain. Therefore, myelin within a GM region of the brain may affect both GMV and WMV estimates in that region. To study morphometric changes in the brain during motor skill learning, we used both GMV and WMV estimates in GM regions and WMV estimates in WM regions. Throughout this article, we will employ GM and WM terms to refer to the classical anatomical regions of the brain and GMV and WMV to refer to the quantitative estimates calculated from the segmented images.

Through longitudinal in vivo sMRI of rodent brain, using a specific MR sequence with a magnetization transfer (MT) pulse to increase contrast between water-rich and water-poor tissue, combined with a cross-sectional immunohistochemical investigation, we describe structural plasticity dynamics during motor skill learning and the associated adaptive cortical myelination. This was studied in wild-type (WT) animals during learning of a skilled, single-pellet forelimb-reaching task. Structural MRI water-rich signal putatively reflects the abundance of myelinated axons, whereas sMRI water-poor signal putatively reflects the scarcity of myelinated axons. Thus, the specific sMRI sequence with MT allows the investigation of changes in putative myelin content in GM and WM regions of the brain. Plasticity in human brain structure has been typically assumed to follow a continuous linear or asymptotic increase throughout the time of training. However, a recent study in humans revealed GM expansion in motor cortex during the first 4 weeks of training followed by partial renormalization (*Wenger et al., 2017b*). To evaluate the fit of these three alternative models to the data, we tested the three corresponding regression models (*Figure 2—figure supplement 2*): (1) a linear model to reflect a continuous formation of candidate circuits, (2) an asymptotic model to reflect the recruitment of new candidate circuits followed by stabilization and, (3) a quadratic model to reflect an initial expansion phase followed by a complete renormalization. In this study, we found that motor learning dynamically modulates macrostructural brain plasticity, identified nonlinear decreases in GMV juxtaposed to nonlinear increases in WMV, and that these changes are associated with adaptive myelination in forelimb sensorimotor cortex.

## Results

### Behavioral improvement after forelimb reach-and-grasp training

A group of mice (n = 39) were trained each day on a forelimb-specific motor learning paradigm, the single-pellet reaching task (SRT) over the course of 15 consecutive days (including 3 pre-training days) and whole-brain structural images were acquired at six time points during the learning paradigm (*Figure 1A–C*, SI Appendix, *Figure 1—video 1*). Successful reaches and accuracy during motor skill training significantly increased with time ($F_{11,364}$ = 25.50, p<0.001 and $F_{11,364}$ = 7.976, p<0.001, respectively) in trained mice (*Figure 1D and E*). The improvement in accuracy was confirmed by an improvement in successful reaches on first attempt throughout the learning paradigm ($F_{11,364}$ = 10.56, p<0.001; *Figure 1F*). A higher level of skill was attained by the group of trained mice compared to a group of nontrained controls (n = 16). The ability of control mice to reach, grasp, and retrieve pellets was measured on the final day of the motor learning paradigm. At experimental day 14, trained animals successfully retrieved 47 ± 3% pellets with accuracy of 18 ± 3% while nontrained control mice performed significantly lower (t = 8.641, df = 45, p<0.001) than trained mice with only 7 ± 2% successful reaches and an average accuracy of 10 ± 2%.

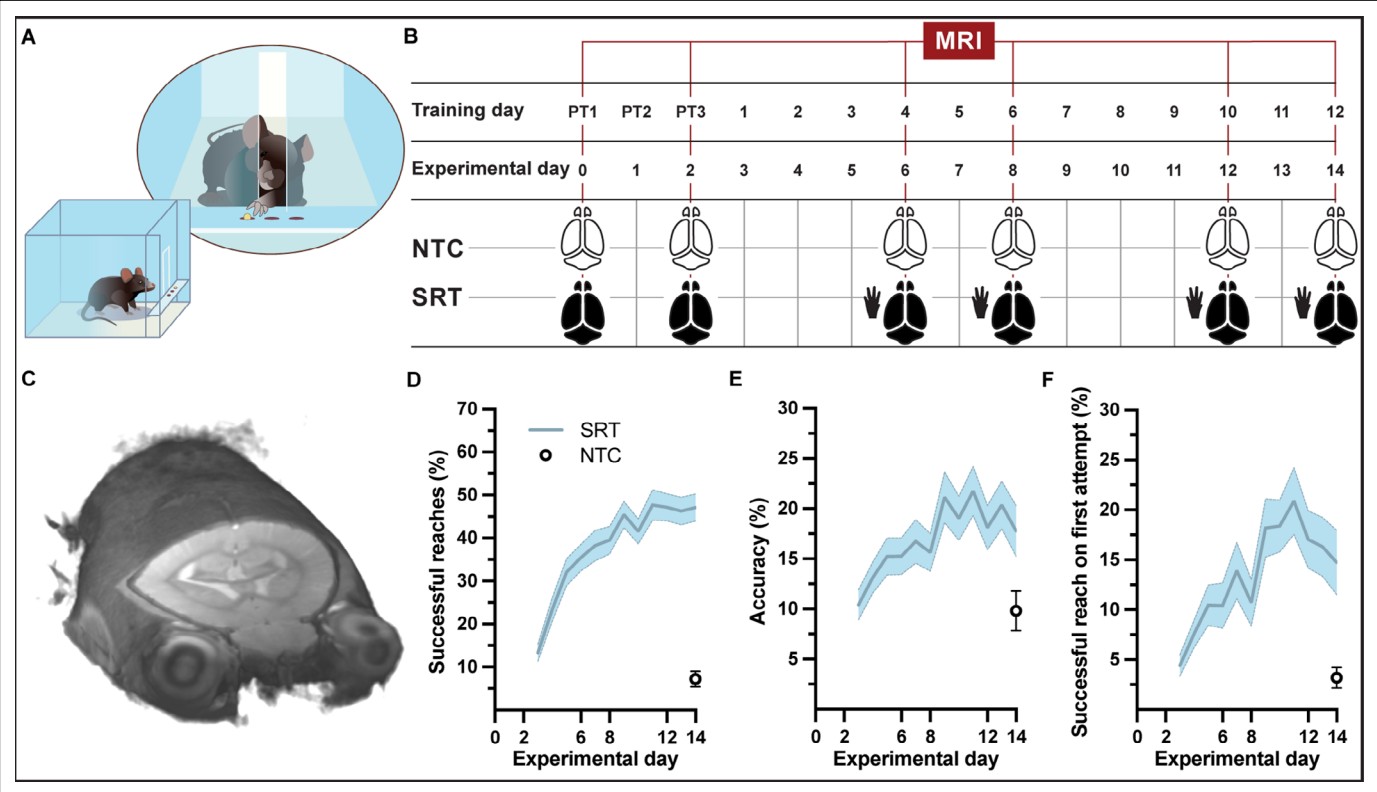

**Figure 1.** Experimental setup and forelimb reach-and-grasp skill learning. (**A**, **B**) Illustration (**A**) and MRI timeline (**B**) during a motor skill behavioral paradigm (SRT: skill reaching trained; NTC: nontrained controls). (**C**) Example of an individual in vivo T1-weighted MRI at 9.4 T at native resolution (0.1 mm isotropic, radiological display). (**D**) Mean performance scores during training of a skilled, single-pellet forelimb reach task, calculated as percentage (47 ± 1) of successful reaches (**D**) and percent (18 ± 1) accuracy (**E**) or percentage of successful reaches on the first attempted reach (19 ± 1) (**F**), during the 12-day training paradigm. Plots (**D–F**) represent the mean and error (SEM) for each performance score of trained subjects; n=37 for experimental days 1-6, n=35 for days 7-8, n=33 for days 9-12, n=31 for days 13-14. Non-trained controls (n=16) were evaluated only at experimental day 14 and the mean and error (SEM) for each performance score is indicated.

The online version of this article includes the following video and figure supplement(s) for figure 1:

**Figure supplement 1.** Mean performance scores of animals used for a cross-sectional myelin immunoreactivity during learning of a skilled, single-pellet forelimb reach task.

**Figure 1—video 1.** A forelimb-specific motor learning paradigm (single-pellet skilled reaching task) providing examples of successful skilled reaches as well as unsuccessful attempts to reach and grasp the pellet.

https://elifesciences.org/articles/77432/figures#fig1video1

An additional group of mice were trained (n = 64 + 8 from the previous group), yet individuals were sacrificed at specific time points during the learning paradigm for cross-sectional analysis of brain tissue. This group of trained mice showed similar improvements in successful reaches and accuracy over the 12 days of training ($F_{11, 336}$ = 36.36, p<0.001 and $F_{11, 336}$ = 5.806, p<0.001, respectively; SI Appendix, *Figure 1—figure supplement 1A and B*). The improvement in accuracy was confirmed by an improvement in successful reaches on first attempt throughout the learning paradigm ($F_{11, 336}$ = 11.67, p<0.001; *Figure 1—figure supplement 1C*). On the final training day, trained animals that completed the 15-day learning paradigm (n = 12) successfully reached-to-grasp an average of 54 ± 5% of pellets with an average accuracy of 22 ± 3%, whereas nontrained controls (n = 3 + 9 from the previous group) exhibited only an average of 7 ± 2% successful reaches with an average accuracy of 11 ± 3%. Skilled reaching performance of trained mice was significantly better than performance of nontrained control animals at the end of the behavioral paradigm (t = 7.425, df = 20, p<0.001), confirming motor skill learning in trained mice.

## Whole-brain structural analysis identified nonlinear decreases in gray matter volume juxtaposed to nonlinear increases in white matter volume during motor learning

We performed in vivo T1-weighted MR imaging at baseline, immediately prior to motor skill training (last day of 'pre-training') and at four time points during motor skill learning ('training'). VBM analyses revealed changes in GMV and WMV in trained and in nontrained control mice (SI Appendix, *Figure 2— figure supplement 1A and B*, *Figure 2—figure supplement 1—source data 1*). To discriminate the effects of motor learning from those of time, the learning effects were evaluated using whole-brain VBM analysis of longitudinal sMRI data on trained mice relative to nontrained controls (i.e., group by time interactions). Three different regression models – linear, asymptotic, and quadratic (SI Appendix, *Figure 2—figure supplement 2*), representing three different time courses – were used and revealed statistically significant changes in both GMV and WMV ($p_{FDR\ corr}$<0.01). Significant nonlinear decreases in GMV relative to the control group were observed juxtaposed with significant nonlinear increases in WMV during learning (*Figure 2A and B*). Interestingly, the asymptotic model provided a much higher number of significant voxels than either the linear or quadratic models and there were no significant linear changes in WM associated with learning (*Table 1*). Areas well-known to be involved in motor learning were identified by VBM analysis, following an asymptotic time-course model in trained animals relative to nontrained controls (SI Appendix, *Figure 2—figure supplement 3*). Significant decreases in GMV and significant increases in WMV were observed in both cortical and subcortical brain areas. In addition, significant MRI signal changes in WMV were observed in many subcortical WM areas (SI Appendix, *Figure 2—figure supplement 4*) using a lower threshold ($p_{FDR\ corr}$<0.05).

Furthermore, whole-brain analysis indicated that, in some brain areas, one model fit the time courses better than the others. Akaike information criterion (AIC) values were used to distinguish these specific areas. Clusters for which AIC values indicated asymptotic modeling revealed GMV decreases in discrete brain areas including primary motor cortex (MOp), primary somatosensory cortex for the forelimb (SSp-ul), and globus pallidus (GPe) (*Figure 2C*). whereas clusters for which AIC values indicated preferred quadratic modeling revealed GMV decreases in the paramedian lobule (PRM) and vermian lobule VI of cerebellum (DEC), superior colliculus (SC), and nucleus accumbens (ACb; *Figure 2D*).

## VBM restricted to cortical sensorimotor areas identified nonlinear decreases in gray matter volume together with nonlinear increases in white matter volume during motor learning

To explicitly investigate changes within cortical regions known to be involved in motor skill learning, based on previous observations, we generated a bilateral atlas-based mask of primary motor cortex (MOp), secondary motor cortex (MOs), and primary somatosensory cortex (SSp) (*Figure 3A and B*). Using this cortical mask, the three different regression models were tested and compared (*Table 2*). This analysis demonstrated that asymptotic modeling was clearly preferred and that statistically significant decreases in GMV ($p_{FDR\ corr}$<0.001) overlapped with significant increases in WMV ($p_{FDR\ corr}$<0.01) in cortical sensorimotor areas in trained animals relative to nontrained controls (*Figure 3A and B*). These were observed in primary motor areas (MOp) and somatosensory cortex for forelimb and hindlimb (SSp-ul and SSp-ll) contralateral to the trained limb, somatosensory barrel field (SSp-bf) ipsilateral to the trained limb, and bilateral secondary motor areas (MOs) and bilateral somatosensory area for the mouth (SSp-m). Additionally, we found GMV decreases in contralateral somatosensory area for the nose (SSp-n) and bilateral GMV decreases in primary motor areas (MOp) and somatosensory cortex for the forelimb and hindlimb (SSp-ul and SSp-ll), as well as in barrel field (SSp-bf). Furthermore, we observed bilateral increases in WMV in primary motor area (MOp), somatosensory cortex for the forelimb and hindlimb (SSp-ul and SSp-ll), somatosensory area for the barrel field (SSp-bf), and somatosensory area for the nose (SSp-n).

To further constrain our analysis, structural data were extracted and analyzed using a nonbiased volume of interest (VOI) for sensorimotor cortex, based on fMRI maps of forepaw stimulation reported by Jung and colleagues in 2019 (*Jung et al., 2019*; *Figure 3C and D*). Nonlinear decreases in GMV and increases in WMV relative to nontrained controls were observed contralateral to the trained limb. These changes followed a nonlinear model rather than a linear one (Δ AICc > 2; *Table 3*). In

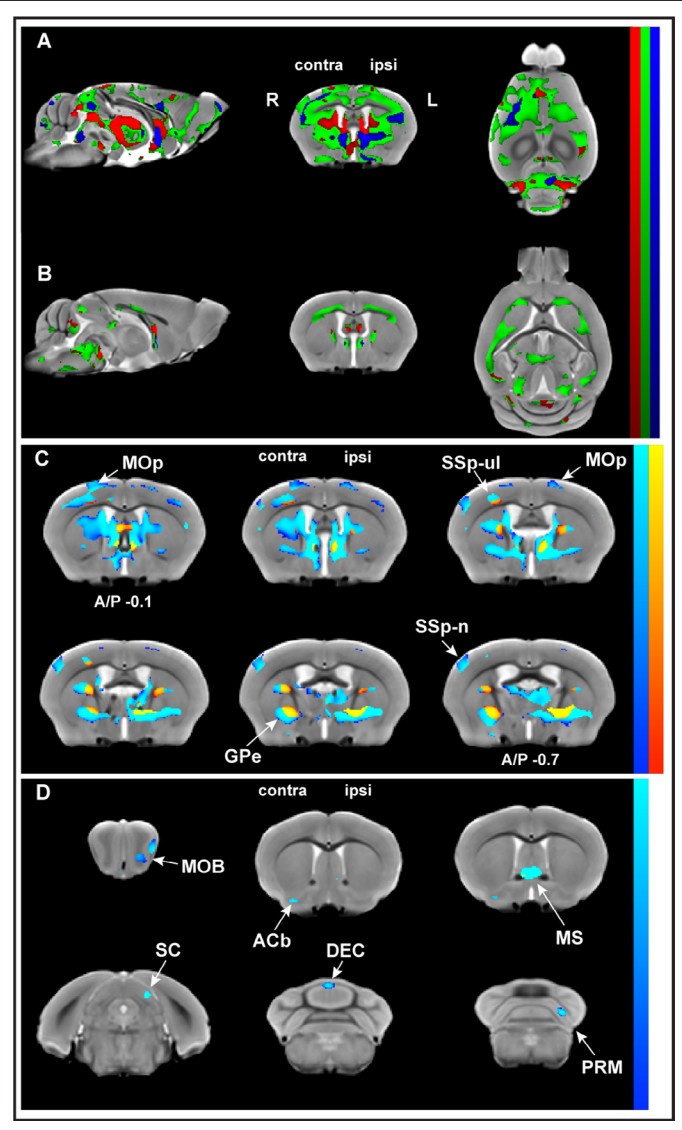

**Figure 2.** Whole-brain structural analysis revealed nonlinear decreases in gray matter volume (GMV) juxtaposed to nonlinear increases in white matter volume (WMV) with learning. Significant changes were observed in GMV (**A**) and WMV (**B**) for which volumetric changes were modeled by three different time courses (linear model in red, asymptotic model in green, and quadratic model in blue) overlayed on the in vivo MRI template created from all subjects in this study. Decreases in volume (cold blue scale) and the increases in volume (warm red scale), in coronal sections ranging from A/P Bregma –0.1 to –0.7 mm, defined using the asymptotic model ($p_{FDR\ corr} < 0.01$) and thresholded at $D_{AIC} > 10$ for asymptotic versus linear and/or quadratic models (**C**). Cortical and subcortical areas following an asymptotic model show decreases in GMV and increases in WMV and include primary motor cortex (MOp), primary somatosensory cortex for the forelimb (SSp-ul), and globus pallidus (GPe) contralateral to the trained limb, among others. Discrete clusters for which Akaike information criterion (AIC) values indicated a preferred quadratic model (**D**). Preferred quadratic clusters are observed in paramedian lobule of cerebellum (PRM), superior colliculus (SC), and main olfactory bulb (MOB) ipsilateral to the trained limb, medial septum and vermian lobule VI (DEC), and nucleus accumbens (ACb) contralateral to the trained limb.

The online version of this article includes the following source data and figure supplement(s) for figure 2:

**Figure supplement 1.** Forelimb reach-and-grasp training dynamically modulates macrostructural brain plasticity.

**Figure supplement 1—source data 1.** Effect of training on gray matter volume (GMV) and white matter volume (WMV) in trained mice and the effect of time in nontrained control mice.

**Figure supplement 2.** Three different regression models representing three different time courses were used to test for different patterns of change in gray matter (GM) and white matter (WM).

*Figure 2 continued on next page*

*Figure 2 continued*

**Figure supplement 3.** Whole-brain structural analysis of nonlinear decreases in gray matter volume (GMV) juxtaposed to nonlinear increases in white matter volume (WMV) with learning.

**Figure supplement 4.** Nonlinear increases in white matter volume (WMV) in subcortical white matter regions of the brain ($p_{FDR\ corr}$<0.05) follow an asymptotic model.

addition, we created two additional VOIs based on known areas of reorganization of forelimb representation using multielectrode recordings and skill reaching (*Tennant et al., 2011*). Structural data were extracted and plotted for the caudal forelimb area (CFA) and the rostral forelimb area (RFA) contralateral to the trained limb (*Table 3*; SI Appendix, *Figure 3—figure supplement 1*) and similar nonlinear changes were observed. Changes in both GMV and WMV followed a quadratic/nonlinear pattern rather than linear (Δ AICc > 2) except for intracortical myelin in RFA where it was not possible to discriminate which model fit best (Δ AICc < 2).

## Motor learning evokes nonlinear plasticity within cortical layers IV–VIa

In order to place the results obtained from cortical sensorimotor-restricted VBM within the different layers of the cortex, we normalized the Allen Mouse Brain Atlas (AMBA) to the in vivo MRI template from this study. The nonlinear decreases in GMV together with nonlinear increases in WMV observed during motor learning in somatosensory cortex span from layer IV through layer VIa (*Figure 4A*). A combined staining of cells and fibers in coronal sections of mouse brain tissue was also used to differentiate and identify cortical layers. VBM results were coregistered to ex vivo histological sections and confirmed observations using the cortical layers from the AMBA (*Figure 4B*). An enlargement of these results is presented in *Figure 4C*. The somatosensory cortex is organized into six layers, much like the rest of the neocortex. Layer 4 is associated with more input, and layer 5 is associated with more output. Thus, the changes observed by VBM involve both cortical input and output of information.

## Motor learning evokes nonlinear plasticity of cortical white matter components that are associated with adaptive myelination

Although we employed a T1-weighted sequence specifically chosen for increased myelin detection within GM, MRI metrics do not provide direct myelin measures. We therefore immunolabeled myelin basic protein (MBP) in coronal brain sections at six different intervals during the learning paradigm (SI Appendix, *Figure 5—figure supplement 1*). These intervals were matched to those used for MRI: baseline, immediately prior to motor skill training, and at four time points during motor skill training. Myelin immunoreactivity was quantified in an area of SSp-ul that presented with a highly significant VBM cluster contralateral to the trained limb, as well as significant changes in ipsilateral SSp-ul when an atlas-based cortical mask was applied to our data (*Figure 5*). We observed a significant correlation

**Table 1.** Training by group interaction effects for gray matter volume (GMV) and white matter volume (WMV).

Whole-brain between-group analysis presenting the significant number of voxels ($p_{FDR\ corr}$<0.01 and <0.001) together with the change in volume (mm$^3$).

| | Changes in GMV | | Changes in WMV | |
|---|---|---|---|---|
| | **Increase** | **Decrease** | **Increase** | **Decrease** |
| **$p_{FDR\ corr}$<0.01** | | | | |
| Linear | - | 45,158 (23.12) | 12,407 (6.35) | - |
| Asymptotic | - | 241,549 (123.68) | 78,442 (40.16) | - |
| Quadratic | - | 28,615 (14.65) | 2250 (1.15) | - |
| **$p_{FDR\ corr}$ <0.001** | | | | |
| Linear | - | 1834 (0.94) | - | - |
| Asymptotic | - | 109,811 (56.22) | 22,095 (11.31) | - |
| Quadratic | - | 2780 (1.42) | 288 (0.15) | - |

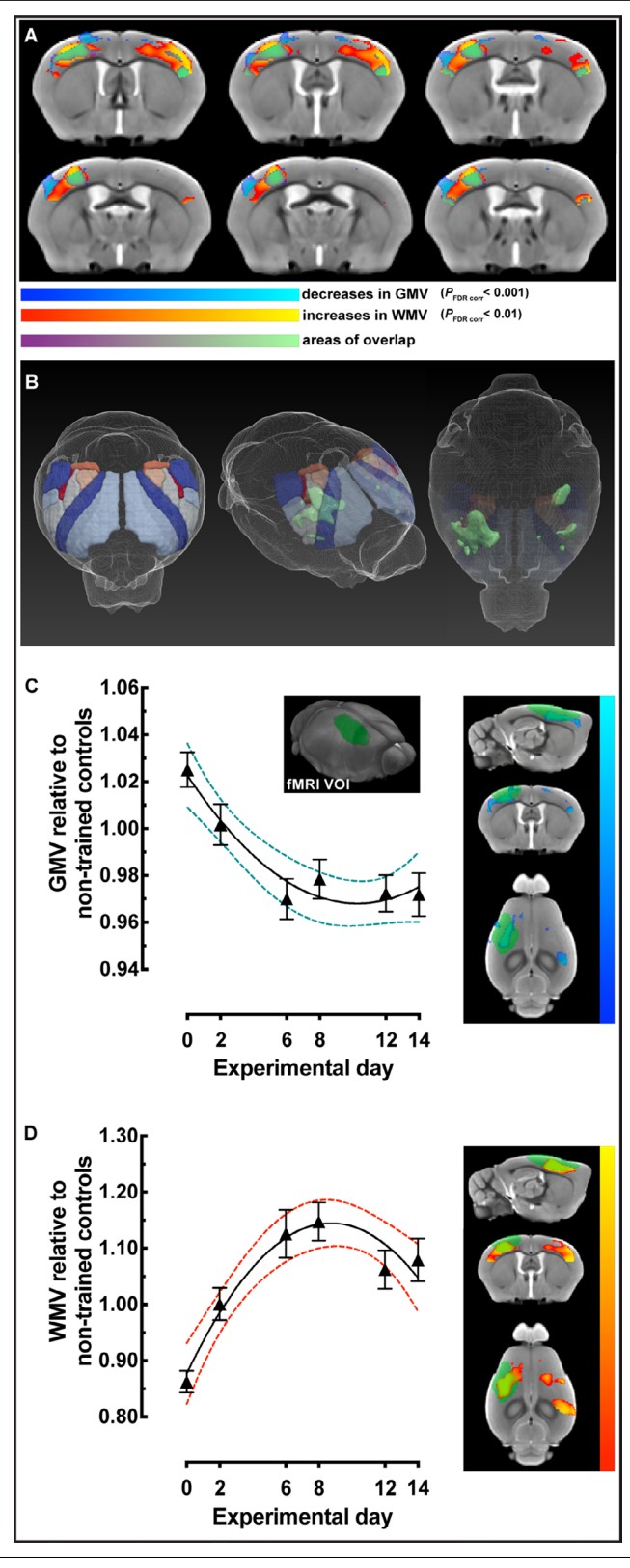

**Figure 3.** Sensorimotor-restricted voxel-based morphometry (VBM) analysis in cortex identified nonlinear decreases in gray matter volume (GMV) together with nonlinear increases in white matter volume (WMV) during motor learning. (**A**) VBM analysis using an atlas-based bilateral mask for primary motor cortex (MOp), secondary motor cortex (MOs), and primary sensory areas (SSp) revealed significant decreases in GMV (cold blue scale) and

*Figure 3 continued on next page*

*Figure 3 continued*

increases in WMV (warm red scale) following an asymptotic model ($p_{FDR\ corr}$<0.001 and <0.01, respectively). Non-linear decreases in GMV are observed together, and overlap (green), with non-linear increases in WMV. (**B**) 3D representations of the atlas-based mask for MOp + MOs + SSp together with the overlap of significant GMV and WMV changes (green). (**C**) GMV changes, relative to nontrained controls, in sensorimotor cortex contralateral to the trained forelimb extracted using a volume of interest (VOI) based on fMRI mapping of forepaw stimulation (VOI represented in green). (**D**) WMV changes, relative to nontrained controls, from the same VOI in (**C**). Plots in (**C**) and (**D**) represent the average extracted value, normalized to non-trained controls (*n*=16), of each VOI for every trained subject ± SEM; *n*=31 at 0, *n*=31 at 2, *n*=33 at 6, *n*=35 at 8, *n*=37 at 10 and *n*=39, at experimental day 14.

The online version of this article includes the following figure supplement(s) for figure 3:

**Figure supplement 1.** Caudal forelimb area (CFA)- and rostral forelimb area (RFA)-restricted voxel-based morphometry (VBM) analysis identified nonlinear decreases in gray matter volume (GMV) together with nonlinear increases in white matter volume (WMV) during motor learning.

---

between WMV, extracted using an unbiased fMRI-based VOI from an independent study (*Jung et al., 2019*), with myelin immunoreactivity in trained animals (*Figure 5B*; Pearson's *r* = 0.75, p=0.03). In line with this result, we found a nonsignificant negative trend between GMV and myelin immunoreactivity (*Figure 5B*; Pearson's *r* = –0.38, p=0.35). The quantification of myelin immunoreactivity within the bilateral cluster located in SSp-ul revealed significant changes with training (one-way ANOVA, $F_{5,66}$ = 2.538, p<0.05; *Figure 5C*). Specifically, myelin immunoreactivity increased up until experimental day 6 after which it began to decrease toward baseline levels. From baseline measurements, at experimental day 0, average MBP immunoreactivity increased by 15% at experimental day 6 followed by an 8% decrease from experimental day 6 to experimental day 14. In line with our observations using VBM, the differences detected in myelin immunoreactivity preferentially followed a nonlinear model rather than linear (Δ AICc > 2) with an 87.8% probability of a preferred nonlinear model compared to a linear model. No significant differences were observed in myelin immunoreactivity between trained animals and nontrained controls at experimental day 14 (*t* = 0.4096, df = 22, p>0.05) (SI Appendix, *Figure 5—figure supplement 2A*) nor in nontrained controls between baseline experimental day 0 and experimental day 14 (SI Appendix, *Figure 5—figure supplement 2B*).

In addition to myelin immunoreactivity, cortical thickness was also quantified in histological sections in sensorimotor cortex where myelin was evaluated (*Figure 5D–F*). No significant difference was observed in cortical thickness between trained mice and nontrained controls (*t* = 0.5561, df = 98, p>0.05, *n* animals = 72 for trained mice, *n* animals = 28 for nontrained controls). Similarly, no difference was found in the cumulative distribution of cortical thickness between groups (Kolmogorov–Smirnov test, p>0.05, *D* = 0.08949, n = 424 for trained mice, n = 191 for nontrained controls).

Morphometric changes in WMV and myelin immunoreactivity in SSp-ul were observed to follow a nonlinear trajectory in which we observed significant increases followed by a total, or partial, return to baseline levels during skill learning. To explore the relationship between learning and adaptive myelination, we evaluated whether learning rate correlates with the asymptotic changes for WMV. We evaluated WMV data extracted from the fMRI VOI contralateral to the trained limb at experimental

---

**Table 2.** Gray matter volume (GMV) and white matter volume (WMV) training by group interaction effects.

Masked cortical areas (MOp + MOs + SSp) between-group analysis presenting the significant number of voxels ($p_{FDR\ corr.}$<0.01) together with the change in volume (mm³).

| | Changes in GMV | | Changes in WMV | |
|---|---|---|---|---|
| | Increase | Decrease | Increase | Decrease |
| $p_{FDR\ corr}$<0.01 | | | | |
| Linear | - | - | - | - |
| Asymptotic | - | 33,817 (17.31) | 13,356 (6.83) | - |
| Quadratic | - | - | - | - |

MOp, primary motor cortex; MOs, secondary motor cortex; SSp, primary somatosensory cortex.

**Table 3.** Comparison of Akaike information criterion (AIC) values for changes in gray and white matter taken from structural data extracted using three unbiased cortical volumes.

| | fMRI VOI | | RFA VOI | | CFA VOI | |
|---|---|---|---|---|---|---|
| | Linear | Quadratic | Linear | Quadratic | Linear | Quadratic |
| **Changes in GMV** | | | | | | |
| AICc | −1271 | −1278 | −1833 | −1835 | −1337 | −1358 |
| Probability of correctness | 3.50% | 96.50% | 24.09% | 75.91% | <0.01% | >99.99% |
| Changes in WMV | | | | | | |
| AICc | −664.3 | −686.6 | −94.80 | −95.33 | −682.2 | −699.8 |
| Probability of correctness | <0.01% | >99.99% | 43.45% | 56.55% | 0.02% | 99.98% |

GMV, gray matter volume; WMV, white matter volume; VOI, volume of interest; RFA, rostral forelimb area; CFA, caudal forelimb area.

day 14 and found a correlation between learning rate and WMV (*Figure 5G*) (Pearson's $r = -0.378$, p=0.0360). Animals in which changes in WMV presented a larger similarity to an asymptotic time course (lower AICc values) exhibited a higher learning rate.

## Confocal microscopy of myelinated axons revealed nonlinear changes in length density with learning

To further investigate the changes detected in myelin immunoreactivity, we used confocal microscopy combined with a quantitative analysis of fibers as previously described (*Hamodeh et al., 2014*) to reconstruct fiber skeletons (*Figure 6A*) and quantitate length density (calculated as length of myelinated fibers per unit tissue volume), diameter and volumetric fraction of myelinated axons (*Figure 6B*). Confocal images were acquired in MBP-immunolabeled coronal sections of trained animals in SSp-ul contralateral to the trained forelimb at baseline, experimental day 6 and experimental day 14 (*Figure 6A*, *Figure 6—figure supplement 1*). There is a significant increase in the length density of myelinated axons from baseline to experimental day 6 followed by a significant decrease toward baseline levels at experimental day 14 (one-way ANOVA, $F_{2,7} = 8.249$, p<0.05; *Figure 6B*). Specifically, the length density of myelinated axons increased 175% from baseline to experimental day 6 and decreased 60% from experimental 6 to experimental day 14. Furthermore, the length density of myelinated axons follows a nonlinear quadratic model ($F_{2,7} = 8.249$, p<0.05; *Figure 6B*) with an $R^2$ of 0.7021. The changes detected in length density of myelinated axons follow a quadratic model rather than a linear one (AIC > 2) with a 99.35% probability of a preferred quadratic model compared to a linear model. Length density is influenced by the remodeling of existing myelin sheaths and the addition of new myelin onto previously unmyelinated regions of axons, either by newly recruited or preexisting oligodendrocytes.

There is a nonsignificant decrease (59%) in myelin sheath diameter from baseline to experimental day 6, followed by a nonsignificant increase (118%) from experimental day 6 to experimental day 14 (one-way ANOVA, $F_{2,7} = 1.196$, p=0.36; *Figure 6C*). Although the changes observed in myelin sheath diameter are substantial, the spread of myelin sheath diameter between subjects at experimental day 0, together with a modest number of observations (n = 4 per experimental day), does not allow for null hypothesis rejection. There was a nonsignificant increase (105%) in myelin sheath volume from baseline to experimental day 6, followed by a nonsignificant increase (25%) in myelin sheath volume from experimental day 6 to experimental day 14 (one-way ANOVA, $F_{2,7} = 1.748$, p=0.24; *Figure 6D*).

## Discussion

Myelination, like synaptic plasticity, contributes to learning by activity-dependent modification of an initially 'hard-wired' circuitry (*Bechler et al., 2018*). The dynamics of myelin plasticity and how it relates to volumetric-based measurements of experience-dependent brain changes remain unknown. In this study, we combined motor skill learning in mice with longitudinal sMRI and immunohistochemistry to study the nature of structural changes that take place in the brain during learning.

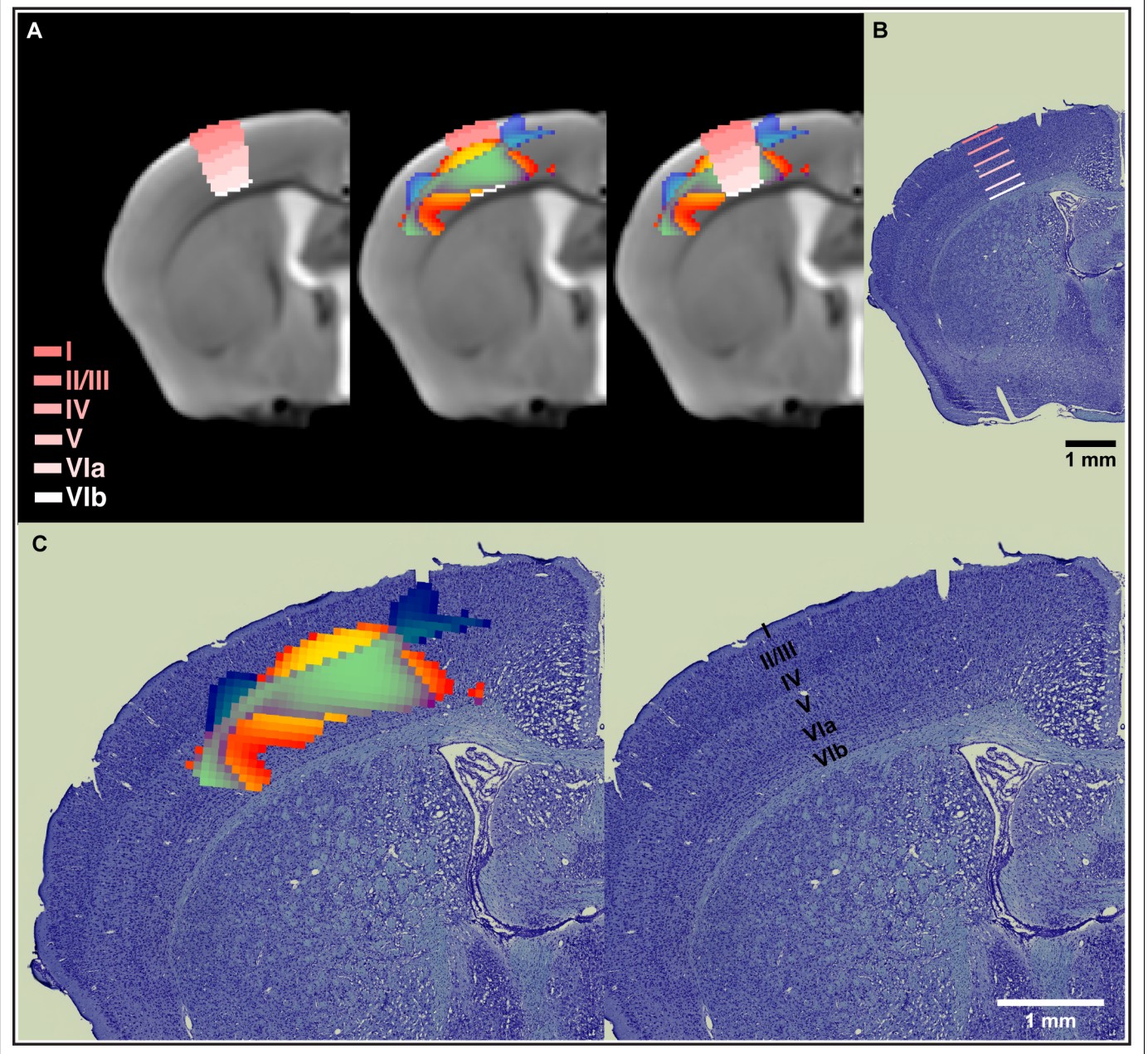

**Figure 4.** Nonlinear decreases in gray matter volume (GMV) together with nonlinear increases in white matter volume (WMV) observed during motor learning in somatosensory cortex span from layer IV through layer VIa. (**A**) Normalization of the Allen Mouse Brain Atlas to the in vivo MRI template from this study allowed for an accurate location of the nonlinear changes in primary somatosensory cortex (SSp) within the different cortical layers. (**B**) The cortical layers disposition from the Allen Mouse Brain Atlas was confirmed using a combined staining of cells and fibers in coronal sections of mouse brain tissue. (**C**) Nonlinear changes in SSp were coregistered to ex vivo histological sections and confirmed observations obtained using the cortical layers from the Allen Mouse Brain Atlas. Nonlinear decreases in GMV (cold blue scale) together with nonlinear increases in WMV (warm red scale) and the overlap of significant GMV and WMV changes (green).

Longitudinal in vivo sMRI acquired throughout learning a skilled, single-pellet forelimb reach task revealed bilateral nonlinear decreases in GMV juxtaposed to nonlinear increases in WMV modeled by an asymptotic time-course function. Specifically, using an atlas-based cortical mask, we found bilateral nonlinear changes with learning in primary and secondary motor areas and in somatosensory cortex. Supporting these results, a cross-sectional analysis unveiled an increase in myelin immunoreactivity in the somatosensory cortex for the forelimb, followed by a return toward baseline. Specifically, using confocal microscopy, we found a significant increase in the length density of myelinated axons from

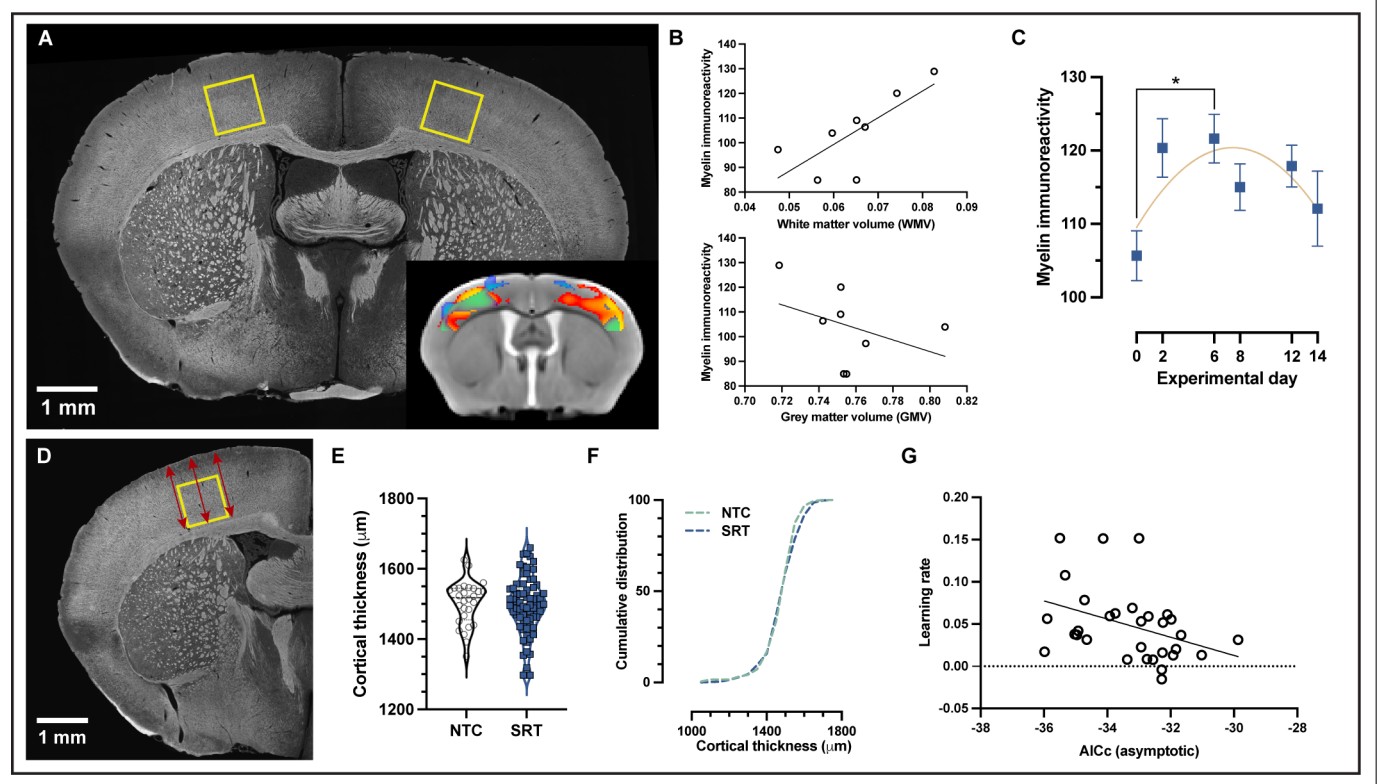

**Figure 5.** Motor learning evokes nonlinear plasticity of cortical white matter components that are associated with structural changes. (**A**) Representative image of myelin immunohistochemistry. (**B**) Positive correlation between cortical white matter volume (WMV) values and myelin immunoreactivity at the individual level in trained animals (Pearson's $r = 0.75$, p=.03) and nonsignificant negative correlation between gray matter volume (GMV) and myelin immunoreactivity (Pearson's $r = –0.38$, p=–.35). (**C**) Myelin immunoreactivity increases until experimental day 6, after which it decreases toward baseline levels. The changes detected in myelin immunoreactivity follow a quadratic model rather than a linear one (AICc > 2). Data are represented as mean ± SEM (n=12 for each experimental day) (**D**) Illustrative representative of the three measurements acquired at the sensorimotor cortex in the same area where myelin was quantified. (**E**) No significant changes were observed in sensorimotor cortical thickness between trained animals and nontrained controls, (unpaired *t*-test; *t*(98)=0.5561) and no differences in the cumulative distribution of cortical thickness were observed between groups in the primary somatosensory cortex for the forelimb (SSp-ul) (**F**). (**G**) Correlation between learning rate and the asymptotic fit (measured as AICc) for WMV on an individual level (Pearson's $r = –0.378$, p=0.0360) in which animals with WMV that best fit an asymptotic model (lower AICc values) exhibited higher learning rates.

The online version of this article includes the following figure supplement(s) for figure 5:

**Figure supplement 1.** Experimental design for the cross-sectional immunohistochemistry analysis of myelin immunoreactivity (**A**) and representative image of myelin basic protein immunohistochemistry (**B**) highlighting the specificity of the immunodetection of myelin in cortex and striatum.

**Figure supplement 2.** Comparison of myelin immunoreactivity between trained and non-trained controls or among non-trained controls.

**Figure supplement 3.** Higher magnification image of a representative myelin basic protein (MBP)-immunolabeled coronal section.

baseline to experimental day 6. Thereafter, the length density of myelinated axons decreases from experimental 6 to experimental day 14. The complementary cortical thickness analysis did not reveal any significant difference between trained animals and nontrained controls. This multimodal approach indicates that nonlinear changes observed in cortical GMV and WMV using VBM are likely caused by changes in tissue composition (e.g., more intracortical myelin, as suggested by the immunodetection analyses) rather than changes in cortical thickness or surface areas of SSp-ul, which are unlikely to happen in the relatively short period of a 15-day learning paradigm. These results are further corroborated by the additional finding of a significant correlation between morphometric WMV and myelin immunoreactivity in the same cortical area. Therefore, WMV calculated from WM-segmented T1-weighted MRI using an MT pulse represents myelin to a substantial degree. Altogether, these observations indicate a nonlinear increase of intracortical myelin with learning. Taking into consideration the observations from confocal microscopy of myelinated axons, we speculate that the nonlinear increase of intracortical myelin observed at experimental day 6 is produced by increased, or de novo,

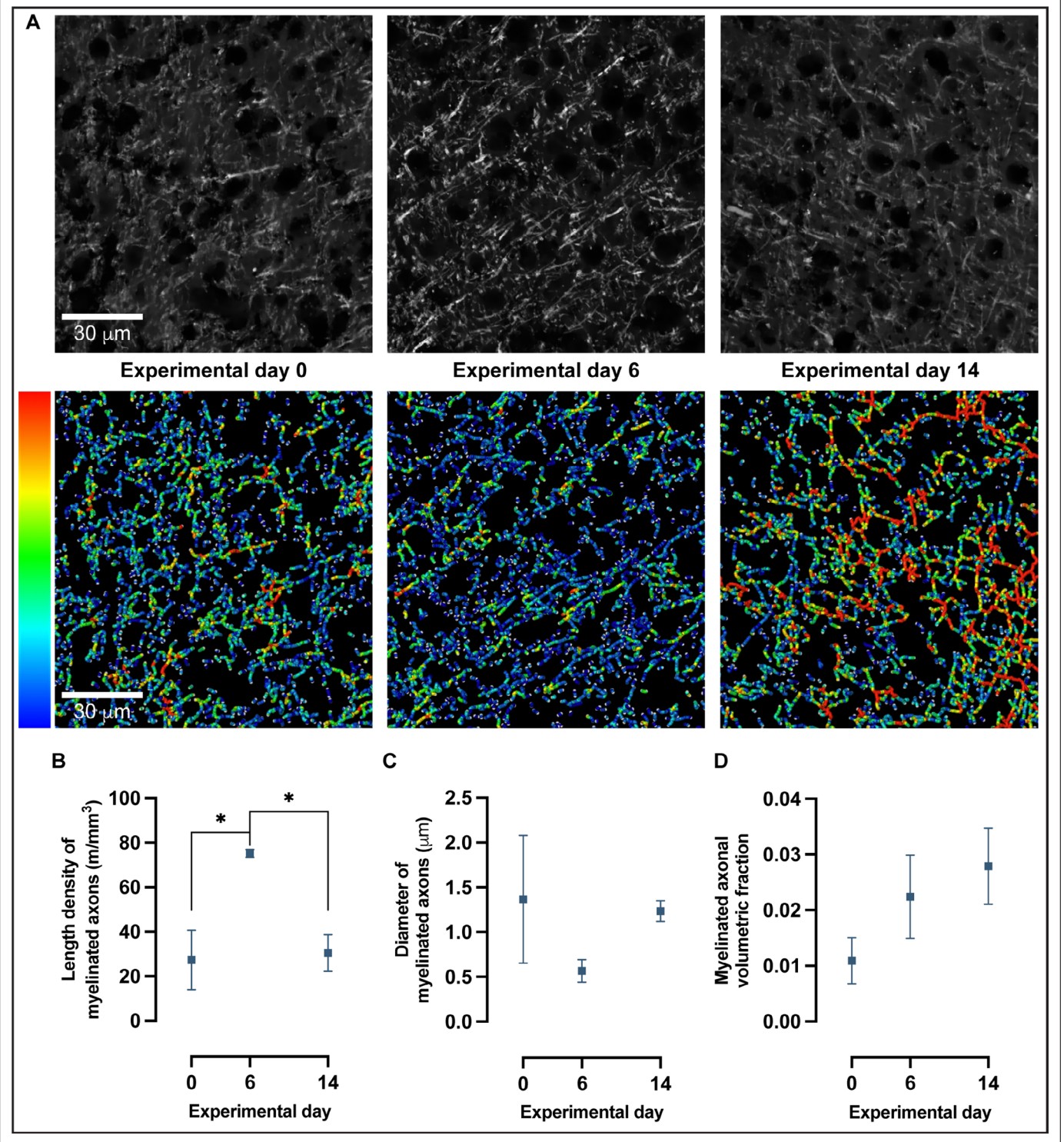

**Figure 6.** Confocal microscopy and 3D reconstruction of myelin basic protein (MBP)-labeled axons revealed nonlinear changes in myelinated axons with motor skill learning. (**A**) Representative confocal images and skeleton reconstructions of myelinated axons at experimental days 0, 6, and 14; pseudo-colorized to reflect thickness. (**B**) Length density of myelinated axons follows a quadratic model rather than a linear one (Akaike information criterion [AIC] > 2) in which there is a 175% increase from baseline to experimental day 6, followed by a 60% decrease from experimental day 6 to experimental day 14 (one-way ANOVA, $F_{2,7}$ = 8.249, $P$ < .05). (**C**) There are no significant changes in diameter of myelinated axons with learning (one-way ANOVA, $F_{2,7}$ = 1.196, alpha level of 0.05) nor in the volumetric fraction of myelinated axons in (**D**) (one-way ANOVA, $F_{2,7}$ = 1.748, alpha level of 0.05). Plotted values represent the mean and error (SEM), $n$=3–4.

*Figure 6 continued on next page*

*Figure 6 continued*

The online version of this article includes the following figure supplement(s) for figure 6:

**Figure supplement 1.** Confocal microscopy evaluation of myelinated axons.

**Figure supplement 2.** Confocal microscopy evaluation of myelinated axons and optical zoom of generated fiber skeletons.

myelination of axons in cortical circuits rather than increases in the diameter of the existing myelinated axons.

Consistent with this idea, oligodendrocyte development has been reported to be required for motor learning in adult mice within the first hours after being introduced to the complex wheel running learning paradigm (*Xiao et al., 2016*). Furthermore, a recent study demonstrated that forelimb-skill reaching dynamically modulates myelination through OPC differentiation (*Bacmeister et al., 2020*). As well, the nonlinear changes observed in somatosensory cortex for the forelimb match with the well-characterized reorganization of forelimb representation during motor skill learning using electrophysiological measurements (*Tennant et al., 2011*).

As previously described in humans (*Wenger et al., 2017b*), and in mice (*Badea et al., 2019*), our in vivo morphometric analysis revealed bilateral changes in several brain regions, including SSp-ul. While structural changes contralateral to the trained forelimb were expected, strong cortical changes ipsilateral to the trained limb are still controversial. Changes in ipsilateral motor cortices are consistent with fMRI findings in humans showing bilateral MOp activation during the execution of a unilateral high-precision motor task (*Buetefisch et al., 2014*). This bilateral activation was later attributed to an inhibitory effect from ipsilateral motor cortex on contralateral motor cortex. Interestingly, this inhibitory effect was modulated by the demand on accuracy of the motor task (*Wischnewski et al., 2016*). Thus, ipsilateral changes in GMV and WMV with learning may be attributed to existing interhemispheric circuits that are strengthened or modified through adaptive myelination. Possible enhanced ipsilateral inhibitory activity may also lead to volumetric changes.

In contrast with our findings of GMV decreases, the study of experience-dependent volumetric changes in primary motor cortex in human adults, trained to write and trace with their nondominant hand, revealed a bilateral GMV expansion followed by a partial renormalization (*Wenger et al., 2017b*). It is possible that the different motor skill learning paradigm used here triggered different structural changes in the brain. On the other hand, these differences could also be attributed to the use of a higher magnetic field in mice than in humans (9.4 T vs. 3 T, respectively) or, perhaps most likely, a consequence of the MRI sequence, optimized for an enhanced detection of myelin. The use of in vivo sMRI in mice using two-session sMRI (before vs. after training) unveiled significant enlargements in GMV in multiple brain regions (*Badea et al., 2019*). In addition, rodent studies using ex vivo DTI with similar motor skill paradigms reported an overall increase in WM in cerebellum (*Badea et al., 2019*) and in corpus callosum directly below primary motor cortex (*Sampaio-Baptista et al., 2013*) after either 23–27 or 11 days of training, respectively. These studies were limited to trained animals versus nontrained control animals at endpoint, and therefore, it is not possible to draw conclusions regarding the temporal dynamics of brain plasticity with learning.

The WMV enlargement we observed in GM by VBM and by myelin immunohistochemistry exhibits an initial expansion, followed by a (partial) renormalization. These observations support the hypothesis that myelin is, at least in part, a component of an expansion-renormalization model comprising exploration, selection, and refinement stages (*Wenger et al., 2017a*; *Makino et al., 2016*; *Lindenberger and Lövdén, 2019*). It is likely that newly formed connections are strengthened in response to repetitive firing of neural circuits produced by a discrete sequence of movements (*Milner et al., 1998*). The increased activity within these circuits likely stimulates additional myelination of involved axons to provide increased efficiency. Experiments utilizing two-photon imaging after inducing monocular deprivation revealed neuron-class-specific myelin plasticity, suggesting that reconfiguration of network connectivity after sensory deprivation requires precise tuning of individual myelination profiles instead of a broad addition of myelin (*Yang et al., 2020*). Thus, the exploration phase of learning involves an expected rise in myelination during initial motor skill improvement, or acquisition. In later stages of motor learning, optimal circuitry is selected to perform the motor task that is subsequently refined through reinforcement and the noneffective candidate circuits are pruned away (*Kilgard, 2012*). Interestingly, once the optimal circuitry is selected and refined, active myelination is

no longer required to support recall and execution of a pre-learned skill (*McKenzie et al., 2014*). Our whole-brain analysis comparing the different possible time-course models identified that the majority of structural changes with learning follow a nonlinear pattern, suggesting that expansion followed by (partial) renormalization is a rather general phenomenon across regions and may be a common principle that unites many manifestations of structural plasticity. Moreover, there is an interestingly large, yet nonsignificant increase in myelin immunoreactivity during the pre-training days, suggesting that pre-training and task familiarization is sufficient to trigger formation of new neuronal connections, stimulating myelination. Consistent with this idea, a previous study in mice showed that the proliferation of OPCs was accelerated in motor cortex within just 4 hr of exposure to the complex wheel. In addition, the introduction to a novel skill learning paradigm stimulated OPC differentiation into newly forming oligodendrocytes after just 2.5 hr of self-training (*Xiao et al., 2016*). Although previous results using the SRT (*Bacmeister et al., 2020*) suggested that production of myelin does not appear until days or weeks later (for review, see *Bonetto et al., 2021*), here we observe that myelination is a rapid and dynamic plastic change throughout learning.

Higher learning rates were associated with a nonlinear/asymptotic model of the changes in the MRI-derived WMV. We could speculate that the selection and refinement phases of the expansion-renormalization model highly influence the proficiency of learning at the individual level. During the exploration phase, a large production of candidate circuits that could potentially elicit effective movements takes place, increasing myelination/myelin levels. However, the magnitude of increase in candidate circuits does not imply a better performance. With a large pool of candidate circuits, the serial activation of the different candidate circuits could lead to behavioral variability (a wide range of diverse performances). Nevertheless, the selection and reinforcement of the circuits that can reliably produce the target movement and the activity-dependent pruning (*Faust et al., 2021*) of the nonefficient candidates to carry out the task lead to a decrease in myelin levels and appear to be related with higher learning rates. Decreasing the number of circuits reduces the behavioral variability and implies better performance. Alternatively, it is possible that the learning-induced reinforcement of selected circuits by myelination contributes to improved communication and oscillatory synchrony within or between distributed neural populations (*Noori et al., 2020*). This plasticity may serve to stabilize oscillatory neural activity across brain regions to facilitate and preserve learned behaviors.

The quadratic changes in myelin immunoreactivity observed by densitometry appear to be influenced by an increase in the length density of myelinated axons followed by a decrease toward baseline levels. The observed increase in length density during the acquisition phase of learning goes in accordance with the idea of an exploration phase of learning associated with the production of substantial number of novel candidate circuits (myelinated axons). These newly formed connections would be lengthy but short in diameter. Afterward, during the consolidation phase of learning, optimal candidate circuits to carry out the task will be selected and refined, for which increases in myelin sheath diameter may occur.

VBM, a method that has been widely used during the past two decades to identify and characterize brain changes among populations, is used to detect systematic density differences of a particular tissue class. In VBM, each voxel in smoothed images contains the average amount of GM around that voxel (*Ashburner and Friston, 2000*). Although modulated GM density (mGM) and cortical thickness are completely different measures, both are commonly used to assess GMV. However, Chung and colleagues found discordances between mGM and cortical thickness analyses (*Chung et al., 2017*). A strong correlation between T1-weighted signal intensity and mGM was reported while no correlation was found between signal intensity and cortical thickness. In cortical VBM, if each voxel is a combination of mGM density, cortical thickness, and surface area, the interpretation of VBM results as volume alone is not entirely correct. Since we did not find significant changes in cortical thickness during learning and that it is unlikely that changes in surface area were produced during the 15-day learning paradigm, we can assume that the morphometric changes observed during learning are consequence of changes in GM and WM density and not volume. Furthermore, this assumption is in line with the changes we observed in myelin immunoreactivity at the histological level.

This study combined longitudinal in vivo MRI with cross-sectional analysis of myelin immunoreactivity in male mice. We limited our experiments to mice with the same sex to focus our study on the learning variable. Some studies incorporating females have indicated differences in certain basic biological processes (e.g., pain processing [*Mogil et al., 2003*] and mechanisms in neuromodulation

[*Huang and Woolley, 2012*]). Thus, by limiting our study to males, any sex differences in the learning process both at the behavioral and structural levels are missing. Future experimental designs should include male and female subjects to identify whether differences exist between sexes.

In this study, we observed the temporal dynamics of experience-dependent macrostructural brain changes during motor skill learning, identified nonlinear decreases in GMV juxtaposed to nonlinear increases in WMV, and found that these changes are associated with adaptive myelination in forelimb sensorimotor cortex. Our results empirically back up the idea that myelination is a rapid initial and partly transient plastic change in learning and supports the use of VBM on WM structural data to evaluate myelinated fibers in cortex.

# Materials and methods

**Key resources table**

| Reagent type (species) or resource | Designation | Source or reference | Identifiers | Additional information |
|---|---|---|---|---|
| Strain, strain background (*Mus musculus*) | C57BL/6J | Jackson Laboratory | #000664 | |
| Antibody | Recombinant anti-myelin basic protein antibody (rat monoclonal) | Abcam | ab7349 | (1:400) |
| Antibody | Alexa Fluor 594 goat anti-rat IgG (H+L) (goat polyclonal) | Jackson ImmunoResearch Laboratories | 112-585-167 | (1:400) |
| Chemical compound, drug | Isoflurane | Piramal Critical Care B.V. | MTnr: 44821 (SE) | |
| Chemical compound, drug | Sodium chloride solution (9 mg/mL) | B.Braun | MTnr: 11054 (SE) | |
| Chemical compound, drug | Viscotears | Bausch & Lomb | MTnr: 12508 (SE) | |
| Chemical compound, drug | L-655,708 | Sigma-Aldrich | L9787 | 0.7 mg/kg |
| Chemical compound, drug | Pentobarbital sodium | Apotek Produktion & Laboratorier | 338327 | 100 mg/kg |
| Chemical compound, drug | Paraformaldehyde | VWR Chemicals | 28794.295 | |
| Chemical compound, drug | Luxol fast blue | Sigma-Aldrich | S3382 | |
| Chemical compound, drug | Cresyl violet | Merck | 1.05235.0025 | |
| Chemical compound, drug | Xylene | VWR Chemicals | 28975.291 | |
| Chemical compound, drug | Glacial acetic acid | Merck | 1.00063.1011 | |
| Chemical compound, drug | Absolute ethanol | VWR Chemicals | 20820.296P | |
| Chemical compound, drug | Lithium carbonate | Merck | Art. 5680 | |
| Chemical compound, drug | O.C.T. compound | VWR Chemicals | 361603E | |
| Chemical compound, drug | Entellan Neu | Merck | 1.07961.0500 | |
| Chemical compound, drug | Mowiol 4-88 | Sigma-Aldrich | 81381 | |

*Continued on next page*

*Continued*

| Reagent type (species) or resource | Designation | Source or reference | Identifiers | Additional information |
|---|---|---|---|---|
| Chemical compound, drug | DABCO | Sigma-Aldrich | D27802 | |
| Software, algorithm | GraphPad Prism | http://www.graphpad.com/ | RRID:SCR_002798 | |
| Software, algorithm | GraphPad Prism | http://www.graphpad.com/ | RRID:SCR_002798 | |
| Software, algorithm | MATLAB | http://www.mathworks.com/products/matlab/ | RRID:SCR_001622 | |
| Software, algorithm | SPM | http://www.fil.ion.ucl.ac.uk/spm/ | RRID:SCR_007037 | |
| Software, algorithm | SPMmouse toolbox | doi:10.1016/j.mri.2013.06.001 | | |
| Software, algorithm | ANTs software package | http://www.picsl.upenn.edu/ANTS/ | RRID:SCR_004757 | |
| Software, algorithm | RStudio (version 4.0.3) | http://www.rstudio.com/ (*R Development Core Team, 2017*) | RRID:SCR_000432 | |
| Software, algorithm | FSL | http://www.fmrib.ox.ac.uk/fsl/ | RRID:SCR_002823 | |
| Software, algorithm | Fiji | http://fiji.sc | RRID:SCR_002285 | |
| Software, algorithm | AMIRA (Advanced 3D Visualization and Volume Modeling) | http://www.fei.com/software/amira-3d-for-life-sciences/ | RRID:SCR_007353 | |
| Software, algorithm | ZEN Digital Imaging for Light Microscopy | http://www.zeiss.com/microscopy/en_us/products/microscope-software/zen.html#introduction | RRID:SCR_013672 | |
| Other | Purified Rodent Tablet | TestDiet | 5TUL | Food pellets used in the motor skill learning paradigm |
| Other | Superfrost plus objective slides | Thermo Scientific (Menzel-Gläser) | J1800AMNZ | Slides used for placement of histological sections |
| Other | Normal goat serum | Jackson ImmunoResearch Laboratories | 005-000-121 | Serum used to prevent nonspecific binding of antibody in immunohistochemistry |

## Experimental design

To study the structural changes that occur in the brain during the acquisition of a novel motor skill, two independent sets of experiments (1 and 2) were performed:

1. motor skilled training was combined with in vivo longitudinal sMRI: trained animals (SRT, n = 39) and nontrained control animals (NTC, n = 16) were scanned at PT1 (baseline), PT3, T4, T6, T10, and T12 (*Figure 1B*). All animals were sacrificed at endpoint (T12).
2. Motor skilled training was performed and animals at different time points during the learning paradigm were sacrificed for a cross-sectional evaluation of myelin immunoreactivity (*Figure 4A*). For the SRT group, 12 animals were sacrificed at PT1, PT3, T4, T6, and T10 and 4 animals at T12. For the NTC group, three animals were sacrificed at T12. Tissue from animals (eight trained animals and nine nontrained controls) from above experiment (1) were also used for evaluation of myelin immunoreactivity. In total, we used 12 animals per time point for cross-sectional evaluation of myelin immunoreactivity.

## Animal care

All procedures were in accordance with protocols approved by the Umeå Regional Ethics Committee for Animal Research (ethical permit: Dnr A 35/2016). A total of 123 young-adult (8- to 11-week-old) male C57BL/6J mice (Jackson Laboratory, Bar Harbor, ME) were used in the study. Animals were housed in a 12 hr/12 hr light–dark cycle under controlled humidity and temperature (23°C). During initial acclimation after delivery, animals were provided food and water ad libitum. All animal handling and behavioral training was carried out during the light phase of the light–dark cycle. Mice were food-restricted 1 week prior to behavioral training. Food restriction was performed gradually to reach

85–90% of their free-feeding weight (calculated by a nonfood-restricted control group). To familiarize mice with the precision pellets (20 mg Purified Rodent Tablet, TestDiet, Richmond, IN) used during motor skill training, 1 g pellets/day were placed into the homecage on the 2 days prior to pre-training. Animal weight was monitored during the entire experiment to ensure that individuals did not fall under 85% of their calculated free-feeding weight on an individual basis.

### Behavioral training: Single-pellet reaching task

A skilled, single-pellet forelimb reach paradigm was performed as previously described in rat (*Molina-Luna et al., 2009*) with some modifications to the training cage (26.5 cm × 9 cm × 20 cm; with grooves to position pellets located 1 cm from inside the cage, see *Figure 1A*). Mice were trained to reach through a narrow slit to grasp and retrieve food pellets positioned within a small indentation located contralateral to the preferred forelimb for each individual animal. Prior to motor skill training, mice were handled and habituated to the behavioral cage during 3 days ('pre-training'; 3 days of 15 min sessions). During the first pre-training day, pellets were placed onto the floor close to the narrow opening at the front of the training cage. During the second and the third pre-training days, pellets were placed onto the shelf located at the front of the cage, outside of the narrow opening and animals occasionally reached to grasp pellets, which was used to determine handedness. During the subsequent 12 days ('training'; 12 consecutive days of 15 min sessions), each animal in the trained group was given a 15 min training session each day that consisted of 30 discrete trials (one pellet/trial). During the entire 15-day experimental paradigm, age-matched nontrained control mice (n = 16) were placed into identical training cages for 15 min and were provided 30 pellets on the cage floor for each experimental day.

### Behavioral analysis

Each of the 12 training sessions were recorded using a digital camera positioned at the front of the training cage. To evaluate motor performance, the number of successful reaches was tallied in addition to the total number of grasping attempts per trial. Video recordings were reviewed if clarification of the score was required. Grasp-to-reach success was calculated as the percent of trials for which food pellets were successfully retrieved from the groove without exhibiting any abnormal behavior (i.e., reaching with the nontrained forelimb or the use of tongue to retrieve to pellet) divided and normalized by the number of trials completed by each individual animal during the 15 min daily training session. Accuracy was calculated as the percentage of successful reaches normalized by the number of attempts performed for each successful trial. The learning rate for each individual was calculated by the slope of a logarithmic model fitted to the learning curve for each individual animal. To evaluate a possible improvement in successful reaches and accuracy over time, restricted maximum likelihood (REML) analysis (allowing for missing values from the animals sacrificed at different time points during the learning paradigm) was calculated for the SRT group. To compare the performance of the SRT group and NTC group, we carried out a t-test analysis on successful reaches on experimental day 14.

### MRI

Animals were scanned and at least 2 hr transpired after waking from anesthesia to behavioral training on any of the MR scan days along the experimental timeline. To prevent any possible isoflurane-induced memory impairment, mice were administered a very low dose (0.7 mg/kg, s.c.) of the highly selective $\alpha_5GABA_A$ receptor inverse agonist, L-655,708 (Sigma-Aldrich, Stockholm, Sweden AB) 15 min prior to the induction of anesthesia (*Saab et al., 2010*). Anesthesia was induced using 4.0% isoflurane mixed with oxygen that was subsequently lowered to 1.5–2% for maintenance during experimental scans. T1-weighted images were acquired using an MT pulse for increased contrast between tissue types with different transfer susceptibilities. We used a T1 3D FLASH sequence (TR/TE = 50/8 ms, flip angle = 20°, using four repetitions) with MT-weighting by Gaussian-shaped off-resonance irradiation (30 μT MT pulse, frequency offset 1.5 kHz, pulse duration 1.8 ms, flip angle 351.2°) performed at 9.4 T (Bruker BioSpec 94/20, running Paravision 6.0 software) with 100 μm isotropic spatial resolution using a 1H Quadrature transmit/receive MRI cryogenic mouse brain RF coil (MRI CryoProbe, Bruker, Germany) for signal reception. The total scan time was 38 min. At the end of each scan, mice were administered saline (10 mL/kg, i.p.) for rehydration and individually placed into a cage to recover from anesthesia before it was returned to its homecage.

## MRI data preprocessing analysis

T1-weighted images were reoriented to match FSL standard orientation convention and were skull stripped using a template-based approach (*Delora et al., 2016*). Skull-stripped images were then bias-corrected for intensity inhomogeneities using the N4 Bias Correction algorithm included in ANTs software package (*Tustison et al., 2010*).

Bias-corrected, skull-stripped brains were manually realigned in SPM8 to approximate the orientation of the stereotaxic, population-averaged, tissue-segmented in vivo brain templates for WT C57Bl/6 mice; described and provided in *Hikishima et al., 2017*. The origin was also set to match the template. The longitudinally acquired scans for each subject (see *Appendix 1—figure 1*) were registered using serial longitudinal registration SPM12 to create an average image for each subject. These averages were then used to create a brain template encompassing all subjects using a serial longitudinal registration of the average from each subject. Our study-specific template was subsequently coregistered and resampled (from 0.1 to 0.08 mm isotropic resolution) to C57Bl/6 template provided in *Hikishima et al., 2017*. Next, the individual scans, from all subjects and time points, were coregistered and resampled to the in vivo study-specific brain template at 0.08 mm isotropic resolution (see *Appendix 1—figure 1*).

A two-stage process was used create our own study- and sequence-specific tissue probability maps (TPMs) based on the GM, WM, and cerebrospinal fluid (CSF) TPMs provided together with the in vivo C57Bl/6 template provided in *Hikishima et al., 2017*. The preprocess segmentation tool from the SPMmouse toolbox (*Sawiak et al., 2013*) (SPM8) was used for the serial longitudinal average of 44 subjects, for which both handedness and training were balanced among the 44 subjects, to create preliminary TPMs from our data. The data were segmented into GM, WM, and CSF images using a mixture of Gaussians and TPMs in SPMmouse. An initial study-specific in vivo brain template was created using the DARTEL toolbox of SPMmouse, which improves registration with an inverse consistent, diffeomorphic transformation. This process was repeated a second time, segmentation followed by DARTEL, but using the preliminary TPMs generated from the first DARTEL step to create our study- and MR sequence-specific TPMs (see *Appendix 1—figure 1*). These final TPMs were then used to segment the individual scans, from all subjects and time points. Modulated and normalized images of GM, WM, and CSF were obtained with DARTEL, multiplied by the Jacobian determinants derived from the spatial normalization. The images were spatially smoothed by convolving with an isotropic Gaussian kernel (full width at half maximum) of five times the voxel size to minimize risk of false positives in statistical analysis.

The individual smoothed and modulated GM and WM TPMs were thresholded at 0.2 (20%) to create GM and WM masks for removal of low-probability voxels. All time point data for each subject were then concatenated into a single 4D image for further statistical analysis.

## VBM statistical analysis

Linear mixed effects (LME) statistical approach was used to test our hypothesis on whether there are significant changes in GM and WM probabilities because of skilled training over time between trained and control groups. LME was chosen as it is capable of handling missing data and enables modeling of random effects in a longitudinal dataset. For this purpose, total intracranial volume (TIV) and amount of training sessions (or time) were defined as fixed effects and subjects defined as random effects for intercept and time to analyze the data. We used R version 4.0.3 (*R Development Core Team, 2017*) with lme4 version 1.1-23 (*Bates et al., 2015*) to perform LME analysis using an in-house coded R script on Ubuntu 18.04.05 LTS workstation. FDR correction was used to correct for multiple comparisons at $p < 0.05$ significance level.

To test for different patterns of change in GM and WM, we used three different regression models (and their opposite function) representing three different time courses: (1) linear, (2) increase followed by a stabilization (inverse-quadratic-asymptotic), and (3) increase followed by a renormalization (inverse-quadratic) as depicted in SI Appendix and *Figure 2—figure supplement 2*. We tested all three regression models for each subject in separate LME analysis to detect changes in GM and WM volumes, both between and within groups.

To study changes in GM and WM with learning specific to cortical areas, we restricted our analysis to M1, M2, and S1 regions using a mask based on the Turone Mouse Brain Template Atlas (TMBTA) (*Barrière et al., 2021*) registered to our in vivo brain template.

## Tissue processing and histology

Animals were anesthetized using 100 mg/kg pentobarbital sodium (i.p., 60 mg/mL, Apotek Produktion & Laboratorier [APL], Kungens Kurva, SE) and transcardially perfused using Tyrode's solution followed by 4% (w/v) paraformaldehyde (PFA) freshly prepared on the same day. After perfusion, brains were post-fixated in 4% PFA at 4°C for 48 hr. Then, PFA was removed and the brains were stored in phosphate buffer (PB) pH 7.40 containing 0.01% (w/v) sodium azide at 4°C. Previous to histology, the brains were cryoprotected in 10% (w/v) sucrose in PB ($Na_2HPO_4 \times 2H_2O$, $Na_2HPO_4 \times H_2O$) with 0.01% (w/v) sodium azide at 4°C. Brains were mounted in O.C.T compound (VWR Chemicals, VWR International, Inc, USA), snap-frozen using high-pressure $CO_2$ and sectioned coronally from Bregma AP 1.70 mm to AP –1.34 mm at 20 µm using a rotatory microtome cryostat (Microm Microtome Cryostat HM 500M). For each brain, at least 10 series of three slides (Superfrost Plus, Thermo Fisher Scientific, Waltham, MA) each containing six sections per slide were obtained. The Kluver–Barrera method (*Kluver and Barrera, 1953*) for the combined staining of cells and fibers in the nervous system was used to differentiate and identify cortical layers in coronal section of mouse brain tissue. Histological sections were imaged using a ZEISS Axioscan 7 slide scanner (Oberkochen, Germany).

## Immunofluorescent staining

Immunodetection of myelin was performed using anti-MBP on coronal sections ranging from Bregma AP –0.1 mm to AP –0.7 mm. Tissue sections were rehydrated in PBS 0.1 M ($Na_2HPO_4 \times 2H_2O$, $Na_2HPO_4 \times H_2O$ + NaCl + KCl) and subsequently blocked using 5% (v/v) goat serum in PBS containing 0.3% Triton X-100 (PBST) for 1 hr at room temperature. Sections were then incubated with rat monoclonal anti-MBP primary antibody (1:400; Abcam, ab7349) in PBST containing 2% (v/v) goat serum for 48 hr at 4°C in a humidified chamber. Sections were washed to remove primary antibody (PBS 0.1 M) and then incubated with fluorescently labeled secondary antibody (Alexa Fluor 594 goat anti-rat IgG (H+L); Jackson ImmunoResearch Laboratories) in PBST containing 2% (v/v) goat serum for 1 hr at room temperature in a humidified chamber. Secondary antibody was removed by washing with PBS 0.1 M and coverslips were mounted using Mowiol 4-88 (Sigma-Aldrich, St. Louis, MO) with 2.5 g/100 mL DABCO (Sigma-Aldrich). Immunolabeled sections were stored at 4° C.

## Image acquisition and quantification

Sections ranging from Bregma AP –0.1 mm to AP –0.7 mm for each animal (n = 84 animals; n = 211 sections) were selected for fluorescence-based microscopy imaging. Images were acquired using a TxRed filter (excitation = 540–580 nm; emission = 600–660 nm) on a Nikon Eclipse Ti-E inverted microscope with a DU897 ANDOR EMCCD camera controlled by Nikon NIS Elements interface, equipped with Nikon CFI Plan Apochromat ×20 (N.A 0.75) objective. Prior to analysis, all the images were aligned using Amira-Avizo Software (version 6.3.0, Thermo Fisher Scientific). A region of interest (ROI) was selected based upon a significant VBM cluster in SSp-ul described in this study. The ROIs were manually positioned and saved for each section using FIJI (*Schindelin et al., 2012*). Signal-to-noise (specific myelin immunoreactivity versus background fluorescence) was determined by the segmentation of each image using FIJI's Multi Otsu Threshold plugin using three different levels of classification. This plugin is based on Otsu's original method but also implements an algorithm described by Liao and Chung (*Liao et al., 2001*). The specific signal was quantified for each section per individual to calculate a mean immunoreactive value for each subject (n = 12 per time point).

## Image acquisition, image preprocessing, and 3D reconstruction of MBP-labeled axons

Sections with Bregma A/P –0.10 mm were selected for confocal imaging (n = 12 animals; n = 12 sections; n = 108 probes). Animals were arbitrarily selected. Experimental days were selected based on VBM and densitometry results. To avoid loss of information, gain parameter ('smart gain') and laser settings were adjusted for an optimal dynamic range [1–255]. Probes were manually located in the area of SSp-ul presenting changes in intracortical myelin. Nine probes per animal were imaged with a laser confocal microscope Leica Sp8 Lia with an HC PL APO ×63/1.40 OIL CS2 objective, using OPSL 552 nm laser to acquire image stacks. Stack of images were acquired using the following parameters: z-stack interval = 0.48 µm; average number of optical sections = 33 (with a minimum of 30 optical sections); matrix size = 1024 × 1024 pixels; voxel xy-size = 0.18 µm × 0.18 µm. The MBP-immunolabeled

axons were 3D reconstructed using a method that was first developed to reconstruct brain blood vessels (*Fouard et al., 2006*) and later adapted to reconstruct neuronal fibers (*Hamodeh et al., 2014*; *Hamodeh et al., 2010*; *Hamodeh et al., 2017*). The fibers are represented by their centerlines with local estimates of their diameters. Fiber skeletons were obtained with the Amira software package (Amira 6.3.1, Konrad-Zuse-Zentrum fur Informationstechnik Berlin [ZIB] and FEI SAS, a part of Thermo Fisher Scientific) and the 'Auto Skeleton' package. This package calculates a distance map of the segmented image and then performs a thinning of the label image by preserving object topology, that is, preventing the splitting of a segment. Prior to applying our segmentation and reconstruction algorithm, preprocessing included 3D Edge-preserving-smoothing using a time stop of 25 and step size of 5 followed by Gaussian filtering using a $5 \times 5 \times 5$ kernel and s = 1. These and following segmentation and reconstruction steps were done with an Amira software package. Segmentation of converted 8-bit gray-level images into a binary image was performed using the Amira 'Auto Thresholding' module with the factorization method and the Otsu criterion (*Otsu, 1979*). The final skeleton structure consisted of myelinated axon segments with cylinders (called nodes) spaced at 0.5 µm along the fibers. The myelinated axonal radius is calculated within the 'Auto Skeleton' function in Amira at each cylinder (or node) between the center of the node and the myelin boundary, taking into account the magnification factor and dimensions of the input data. Amira skeleton was exported to MATLAB as x,y,z point coordinates with respective fiber diameter as csv file. Only those fiber-reconstructed points were used that were within a 'well'-labeled region of the section. Due to antibody penetration issue, only a part of the section is stained depending on tissue depth. Therefore, it is important to take the amount of labeling as a function of section depth into account before using the data for quantification. We systematically limited our fiber of analysis to section depths that had a 0.99 quantile of more than 75 (out of 255). This section depth (Thref) was used to calculate the densities within the probes. Only probes that allowed a sufficient depth (>2 µm) of such staining were used in the quantification. Finally, we added a 0.5 µm section depth correction to account for the structures that bordered on the probe limits (*Abercrombie, 1946*). The average depth that we obtained was 3.6 µm (std = 2, n = 93).

## Cortical thickness analysis

Cortical thickness (layers I–VI) was measured in SSp-ul at the location of the ROI used to quantify myelin immunoreactivity. Three measurements were made in both hemispheres for each animal corresponding to the area of the significant cluster observed from whole-brain VBM analysis. The measurements were acquired from the length of three lines that were drawn based on the ROIs for myelin immunoreactivity. Data were from 60 trained animals and 24 nontrained controls. In total, 424 measurements from trained animals and 191 measurements from nontrained animals were used. Data from any images in which cortex was partially damaged, confounding a proper measurement, were not included.

## Statistical analysis

Student's *t*-test was used to compare two groups and one-way ANOVA with Tukey's test for multiple comparisons. Akaike's information criterion was used to calculate the goodness of fit. The probability of correctness for a model was computed using the next equation: probability = $e^{0.5D}/1+e^{0.5D}$, where D is the difference between the AICc values. Figure legends specify the statistical test used in each case and the number of independent measurements (*n*) evaluated. Behavioral improvement, myelin immunoreactivity mean intensities, and cortical thickness were analyzed using Prism 9.0.0 for macOS (GraphPad Software, San Diego, CA).

## Acknowledgements

This work was supported by Umeå University Medical Faculty, Umeå Sweden (DM); StratNeuro, Umeå University, Umeå Sweden (DM); Swedish Research Council, Stockholm, Sweden (Grant 2018-01047) (ML); Kempe Foundation, Örnsköldsvik, Sweden (grant JCK-1922.2) (DM, FS); *Insamlingsstiftelsen för medicinsk forskning* 2019 (DM); *Magnus Bergvalls Stiftelse*, Stockholm, Sweden (grant 2016-01639) (DM); Swedish Research Council, Stockholm, Sweden 2015-01717 (CB) and 2018-01047 (ML); Agence Nationale pour la Recherche (ANR-16-CE28-0008-01) (CB). HME-S was supported by a grant from the *Agencia Canaria de Investigación, Innovación y Sociedad de la Información* (ACIISI) of the *Regional Consejería de Economía, Industria y Comercio*, Canary Islands Government and European Social Fund

(Canarias 2014–2020, Axis 3 Priority Theme 74 (85%)). We also would like to acknowledge the Small Animal Research and Imaging Facility (SARIF) at Umeå University for providing the MRI equipment to perform the study and the Biochemical Imaging Center at Umeå University (BICU) and National Microscopy Infrastructure, NMI (VR-RFI 2019-00217) for providing assistance in microscopy. With great appreciation, we also thank Dr Seong-Gi Kim and Dr Won Beom Jung for providing the mask of the fMRI-based sensorimotor cluster reported in Jung et al., NeuroImage, 2019.

## Additional information

### Funding

| Funder | Grant reference number | Author |
|--------|------------------------|--------|
| StratNeuro program at Umeå University | | Daniel J Marcellino |
| Kempestiftelserna | JCK-1922.2 | Daniel J Marcellino Fahad R Sultan |
| Insamlingsstiftelsen för medicinsk forskning Umeå Universitet | | Daniel J Marcellino |
| Magnus Bergvalls Stiftelse | 2016-01639 | Daniel J Marcellino |
| Vetenskapsrådet | 2015-01717 | Claudio Brozzoli |
| Agence Nationale de la Recherche | ANR-16-CE28-0008-01 | Claudio Brozzoli |
| Agencia Canaria de Investigación, Innovación y Sociedad de la Información | | Héctor M Estévez-Silva |
| Umeå University Medical Faculty | | Daniel J Marcellino |
| Vetenskapsrådet | 2018-01047 | Martin Lövdén |

The funders had no role in study design, data collection and interpretation, or the decision to submit the work for publication.

### Author contributions

Tomas Mediavilla, Formal analysis, Investigation, Methodology, Writing – original draft, Writing – review and editing; Özgün Özalay, Data curation, Software, Formal analysis, Writing – review and editing; Héctor M Estévez-Silva, Formal analysis, Investigation, Writing – review and editing; Bárbara Frias, Investigation, Writing – review and editing; Greger Orädd, Formal analysis, Methodology, Writing – review and editing; Fahad R Sultan, Resources, Methodology; Claudio Brozzoli, Conceptualization, Methodology, Writing – review and editing; Benjamín Garzón, Conceptualization, Resources, Software, Formal analysis, Funding acquisition, Methodology, Writing – review and editing; Martin Lövdén, Conceptualization, Supervision, Funding acquisition, Methodology, Writing – review and editing; Daniel J Marcellino, Conceptualization, Formal analysis, Supervision, Funding acquisition, Investigation, Visualization, Methodology, Writing – original draft, Project administration, Writing – review and editing

### Author ORCIDs

Daniel J Marcellino http://orcid.org/0000-0002-4618-7267

### Ethics

All procedures involving animals were in accordance with protocols approved by the Umeå Regional Ethics Committee for Animal Research (ethical permit: Dnr A 35/2016).

### Decision letter and Author response

Decision letter https://doi.org/10.7554/eLife.77432.sa1
Author response https://doi.org/10.7554/eLife.77432.sa2

# Additional files

## Supplementary files
• Transparent reporting form

## Data availability

The structural MRI raw data and the scripts employed for the image processing and analysis have been uploaded to Dryad. All these files are provided together with an explanatory document that would allow to reproduce our results.

The following dataset was generated:

| Author(s) | Year | Dataset title | Dataset URL | Database and Identifier |
|---|---|---|---|---|
| Marcellino D, Mediavilla T | 2022 | Skilled Reaching Structural MRI | https://dx.doi.org/10.5061/dryad.crjdfn36c | Dryad Digital Repository, 10.5061/dryad.crjdfn36c |

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

## Appendix 1

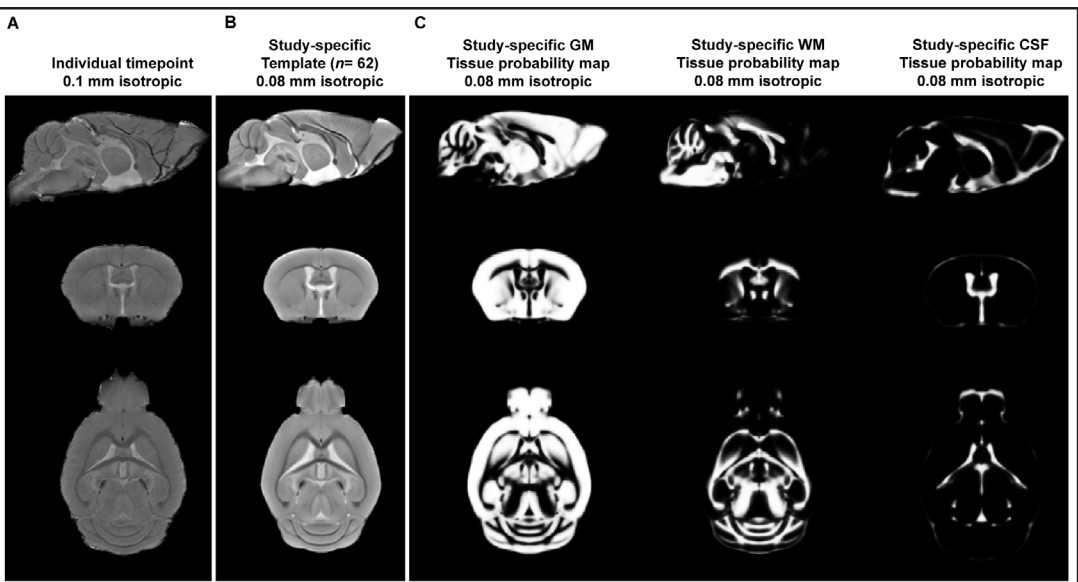

**Appendix 1—figure 1.** Figure depicting sagittal, coronal, and horizontal sections of an in vivo T1-weighted image from one individual scan during the longitudinal study (**A**) and the study-specific template created for mouse brain (**B**). Study- and MR sequence-specific brain tissue probability maps (**C**). Sagittal, coronal, and horizontal sections of tissue probability maps (TPMs) of gray matter (GM), white matter (WM), and cerebrospinal fluid (CSF).

