## [Editor Report]

This study is a convincing and useful addition to the literature on the role of adaptive myelination during fine-motor learning, addressing the important question of whether learning is associated with cortical structural changes as assessed by longitudinal magnetic resonance imaging (MRI) measurements in mice. Novel findings include the observation that MRI measures of myelination increase rapidly in the early stages of motor learning and then decrease during later stages, consistent with models of learning in which initial rapid neural circuit modification is followed by stabilization and pruning. The authors show that a more direct measure of myelination – myelin basic protein immunoreactivity – also follows this non-linear type of time course. In demonstrating these learning-related changes, the study also increases confidence that MRI-based metrics can be used to follow myelin changes non-invasively in "real-time".

---

## [Decision Letter]

**Decision letter after peer review:**

Thank you for submitting your article "Learning-related contraction of grey matter in rodent sensorimotor cortex is associated with adaptive myelination" for consideration by *eLife*. Your article has been reviewed by 2 peer reviewers, and the evaluation has been overseen by a Reviewing Editor and Kate Wassum as the Senior Editor. The reviewers have opted to remain anonymous.

Essential revisions:

1) As indicated in the reviews that stress the importance of demonstrating changes in myelin sheaths, quantitative myelin immunolabelling is a crucial piece of supporting evidence for the suggestion that MRI signal changes reflect adaptive myelination. Please show higher-resolution images of the MBP immunolabeling. As indicated in the reviews, please also provide a more detailed histological analysis of myelin sheaths to complement and reinforce the densitometry.

2) The authors use the terms grey and white matter for specific MRI signals designed to detect relatively water-rich or water-poor domains that are presumed to reflect the abundance of myelinated versus unmyelinated fibers, but not necessarily the classic anatomical grey or white matter, which is confusing. Please better explain this terminology (and possibly even change the terminology) as suggested by the reviewers.

3) Please show some MR images at greater magnification to make it more apparent to the reader within which cortical layers there are increases and decreases in the specified volumes.

4) A discussion/explanation should be provided addressing the issue of why the authors did not find significant MRI signal changes in subcortical white matter or if this white matter was excluded from the analysis.

5) Please explain the rationale for the three models they used to analyze the data.

6) Please make clear that male mice were the subjects in the abstract and discuss the limitation of the exclusion of females.

7) If you have not already done so, please include a key resource table.

*Reviewer #2 (Recommendations for the authors):*

At the same time, the authors should use more detailed histology to explain the reciprocal changes in GM and WM (but see below as regards their terminology). Does GM decrease simply because of an increase in myelin sheath numbers within the same volume, or is there a genuine reduction in neurons/process volume? This may require electron microscopy, but it is an important part of the work.

[Editors' note: further revisions were suggested prior to acceptance, as described below.]

Thank you for resubmitting your work entitled "Learning-related contraction of grey matter in rodent sensorimotor cortex is associated with adaptive myelination" for further consideration by *eLife*. Your revised article has been evaluated by Kate Wassum (Senior Editor) and a Reviewing Editor.

The manuscript has been improved but there are some remaining issues that need to be addressed, as outlined below:

Revisions Required:

1) When the term "non-linear change" is first introduced, it would be helpful to explain that it means non-linear with respect to time from the start of the learning paradigm.

2) The term "length density" could also be defined on first use. Presumably it means length of myelin sheath per unit tissue volume. It could also be explained that this does not distinguish between elongation of existing myelin sheaths and addition of new myelin on previously unmyelinated regions of axons, either by newly generated or pre-existing oligodendrocytes.

3) How is myelin sheath thickness determined from the data illustrated in Figure 6? If it is from direct (or automated) measurement of sheath diameters in high-mag confocal images of MBP-labelled sections, then some representative high-mag images should be shown.

4) In the Discussion (e.g., line 426) "formation of new cortical circuits" could perhaps be better described as "increased or de novo myelination of axons in cortical circuits" since the neuronal circuits/synaptic connections are presumably already in place at the start of training? Similarly (line 444) "…existing …connectivity that is enhanced and re-wired…" might be better: "...existing.. circuits that are strengthened or modified through adaptive myelination."

5) The Discussion of adaptive myelination is framed in terms of wiring or re-wiring particular circuits that drive specific aspects of motor behaviour, rather like the wiring diagram of an electronic appliance (e.g., lines 471-473). However, another way of thinking about the problem is in terms of the potential effects of myelination on synchronization of bulk rhythmic activity within or between brain regions. Maybe this alternative view could also be mentioned? Arguably, it might be more compatible with the sort of macrostructural alterations that are observable using MRI?

*Reviewer #1 (Recommendations for the authors):*

The authors have responded to previous reviews by e.g. clarifying their use of the terms GMV and WMV within the cortex. The m/s is a clear and useful addition to the literature on the role of adaptive myelination during fine-motor learning. Novel findings include the observation that MRI measures of myelination increase rapidly in the early stages of learning and decrease again during later stages, consistent with models of learning in which initial rapid circuit modification is followed by stabilization and pruning. The authors show that a more direct measure of myelination – MBP immunoreactivity – also follows this non-linear type of time-course. In demonstrating these learning-related changes the study also increases confidence that MRI-based metrics can be used to follow myelin changes non-invasively in "real time".

---

## [Author Response]

Essential revisions:1) As indicated in the reviews that stress the importance of demonstrating changes in myelin sheaths, quantitative myelin immunolabelling is a crucial piece of supporting evidence for the suggestion that MRI signal changes reflect adaptive myelination. Please show higher-resolution images of the MBP immunolabeling. As indicated in the reviews, please also provide a more detailed histological analysis of myelin sheaths to complement and reinforce the densitometry.

In the revised version of the manuscript, we now present higher-magnification of the images that were used to quantify MBP immunoreactivity (densitometry) (see Main Figure 5-Supplementary Figure 3 in the revised version of the manuscript). In addition, new immunohistochemical experiments were performed and a second method was used to investigate myelinated axons within the cortex. Coronal sections were immunolabeled for myelin basic protein (MBP) and high-resolution confocal imaging was performed on a subset of trained mice (n=12 mice, n=108 probes, 9 probes per animal, represented in Main Figure 6-Supplementary Figure 1 in the revised version of the manuscript). We acquired Z-stacks with a minimum of 30 optical sections and performed an analysis of fibers based on a quantitative 3D immunohistochemical method (3D-QICH) to reconstruct and analyze length density, diameter and volumetric fraction of myelinated axons. This method of analysis of fibers was first implemented to measure brain vascularity (Fouard *et al.*, 2006) which was later developed further and validated for the systematic analysis of axons (Hamodeh *et al.*, 2010; Hamodeh *et al.*,2014; Hamodeh *et al.*, 2017). The method employed for the 3D-reconstruction and analysis of myelinated axons is explained in detail in the Material and Methods section of the revised manuscript. There is a significant increase in the length density of myelinated axons from baseline to experimental day 6 followed by a significant decrease towards baseline levels at experimental day 14 (one-way ANOVA, F2,7 = 8.249, P <.05; Main Figure 6B), following a quadratic model rather than a linear one (AIC > 2).

Fouard, C., Malandain, G., Prohaska, S., & Westerhoff, M. (2006). Blockwise processing applied to brain microvascular network study. IEEE Trans.Med Imaging, 25(10), 1319-1328.

Hamodeh, S., Eicke, D., Napper, R. M. A., Harvey, R. J., & Sultan, F. (2010). Population based quantification of dendrites: evidence for the lack of microtubule-associate protein 2a,b in Purkinje cell spiny dendrites. Neuroscience, 170(4), 1004-1014. doi:10.1016/j.neuroscience.2010.08.021

Hamodeh, S., Sugihara, I., Baizer, J., & Sultan, F. (2014). Systematic analysis of neuronal wiring of the rodent deep cerebellar nuclei reveals differences reflecting adaptations at the neuronal circuit and internuclear level. J Comp Neurol, 522, 2481-2497.

Hamodeh, S., Bozkurt, A., Mao, H., & Sultan, F. (2017). Uncovering specific changes in network wiring underlying the primate cerebrotype. Brain Struct Funct, 222(7), 3255-3266. doi:10.1007/s00429-017-1402-6

2) The authors use the terms grey and white matter for specific MRI signals designed to detect relatively water-rich or water-poor domains that are presumed to reflect the abundance of myelinated versus unmyelinated fibers, but not necessarily the classic anatomical grey or white matter, which is confusing. Please better explain this terminology (and possibly even change the terminology) as suggested by the reviewers.

In the revised version of the manuscript, we have addressed this issue. We first briefly described within the Introduction the technique of *Voxel-based morphometry* (*VBM*) and defined the terminology employed throughout the manuscript. Specifically, we employed grey matter (GM) and white matter (WM) terms to refer to the classical anatomical regions of the brain and GMV and WMV to refer to the quantitative estimates calculated from the segmented images. We revised and changed the terminology along the manuscript to make it clearer to the reader.

3) Please show some MR images at greater magnification to make it more apparent to the reader within which cortical layers there are increases and decreases in the specified volumes.

In the revised version of the manuscript, we have now included our results on MR images at greater magnification together with a representation of the cortical layers that are delineated by the Allen Mouse Brain Atlas (AMBA). We also performed a combined histological staining of cells and fibers in coronal sections of mouse brain tissue to differentiate and identify cortical layers. VBM results were coregistered to histological sections to clarify in which cortical layers increases and decreases are observed. These are presented in Main Figure 4 of the revised version of the manuscript.

4) A discussion/explanation should be provided addressing the issue of why the authors did not find significant MRI signal changes in subcortical white matter or if this white matter was excluded from the analysis.

VBM was used to investigate changes in grey matter volume (GMV) and white matter volume (WMV). For whole-brain analyses, all subcortical white matter regions were included in the analysis of WMV. Table 1 in the revised version of the manuscript indicate the significant changes and the direction of these changes: decreases in GMV (Main Figure 2A) and increases in WMV (Main Figure 2B). Significant changes were found in WMV, but these were not represented in the Figures originally presented. Instead, we chose to depict significant changes at *P*_FDR corr_ < 0.01 for increases in WMV and *P*_FDR corr_ < 0.001 for decreases in GMV, due to the high number of significant voxels at *P*_FDR corr_ < 0.05, for both WMV and GMV. Figure 2-Supplementary Figure 4 depicts significant increases in WMV according to the asymptotic model at *P*_FDR corr_ < 0.05. Clear changes are observed in subcortical WMV, however, we chose to present higher thresholded results (*P*_FDR corr_ < 0.01) to present the more discrete clusters of increases in WMV together with the more discrete clusters of decreases in GMV at *P*_FDR corr_ < 0.001.

Nevertheless, VBM analysis may present limitations to identify increases in WMV within highly myelinated areas. In white matter regions of the brain, the probability of each voxel to be classified as WM during the segmentation of a T1w image is very large (close to 1), constricting the ability to identify increases in WMV since the probability of a voxel to be classified as WM cannot be higher than 1. This reasoning also can be applied to the significant increases in WMV we observed within GM areas of the brain, for which we interpreted these results as increases in myelin within GM regions of the brain. The use VBM on WM probability maps has only been used, to the best of our knowledge, to characterize decreases in WMV in ALS (Steinbach *et al.*, 2020) and chronic fatigue syndrome (Finkelmeyer *et al.*, 2018).

Steinbach, R., Batyrbekova, M., Gaur, N., Voss, A., Stubendorff, B., Mayer, T., Gaser, C., Witte, O., Prell, T. and Grosskreutz, J., 2020. Applying the D50 disease progression model to gray and white matter pathology in amyotrophic lateral sclerosis. NeuroImage: Clinical, 25, p.102094.

Finkelmeyer, A., He, J., Maclachlan, L., Watson, S., Gallagher, P., Newton, J. and Blamire, A., 2018. Grey and white matter differences in Chronic Fatigue Syndrome – A voxel-based morphometry study. NeuroImage: Clinical, 17, pp.24-30.

5) Please explain the rationale for the three models they used to analyze the data.

In the revised version of the manuscript, we have added the text (below) to the end of the introduction.

“To evaluate the fit of these three alternative models to the data, we tested the three corresponding regression models (Main Figure 2-Supplementary Figure 2): (*i)* a linear model to reflect a continuous formation of candidate circuits, (*ii)* an asymptotic model to reflect the recruitment of new candidate circuits followed by stabilization and, (*iii)* a quadratic model to reflect an initial expansion phase followed by a complete renormalization.”

6) Please make clear that male mice were the subjects in the abstract and discuss the limitation of the exclusion of females.

Since only male mice were used in the study, we have specified this within the Abstract for clarity and have added the text (below) to the Discussion in the revised version of the manuscript.

“This study combined longitudinal in vivo MRI with cross-sectional analysis of myelin immunoreactivity in male mice. We limited our experiments to mice with the same sex to focus our study on the learning variable. Some studies incorporating females have indicated differences in certain basic biological processes; (*e.g.*, pain processing (Mogil *et al.*, 2003) and mechanisms in neuromodulation (Huang *et al.*, 2012)). Thus, by limiting our study to males, any sex differences in the learning process both at the behavioral and structural levels are missing. Future experimental designs should include male and female subjects to identify whether differences exist between sexes.”

Mogil JS, Wilson SG, Chesler EJ, Rankin AL, Nemmani KVS, Lariviere WR, Groce MK, Wallace MR, Kaplan L, Staud R, et al.: The melanocortin-1 receptor gene mediates female-specific mechanisms of analgesia in mice and humans. Proc Natl Acad Sci U S A 2003, 100:4867–4872.

Huang GZ, Woolley CS: Estradiol Acutely Suppresses Inhibition in the Hippocampus through a Sex-Specific Endocannabinoid and mGluR-Dependent Mechanism. Neuron 2012, 74:801–808.

7) If you have not already done so, please include a key resource table.

A key resource table is now included in the revised version of the manuscript.

Reviewer #2 (Recommendations for the authors):At the same time, the authors should use more detailed histology to explain the reciprocal changes in GM and WM (but see below as regards their terminology). Does GM decrease simply because of an increase in myelin sheath numbers within the same volume, or is there a genuine reduction in neurons/process volume? This may require electron microscopy, but it is an important part of the work.

We thank this reviewer and agree that our choice of terminology was, indeed, confusing. Following this Reviewer suggestion, we adjusted our terminology and employed grey matter (GM) and white matter (WM) terms to refer to the classical anatomical regions of the brain and grey matter volume (GMV) and white matte volume (WMV) to refer to the quantitative estimates calculated from the segmented images. In the revised version of the manuscript we have described the terminology within the Introduction and adapted the terminology throughout the manuscript. Our interpretation of the reciprocal changes in GMV and WMV is indeed due in a large part to an increase in myelin sheath number and length density of myelinated axons (within the same volume). We have included an additional series of evaluation of myelinated axons in somatosensory cortex that is presented in Main Figure 6 in the revised manuscript. The results further support adaptive changes in intracortical myelin during learning. We understand that electron microscopy would be an excellent technique to demonstrate changes at the ultrastructural level. Unfortunately, the tissue available from our experiments was not processed for EM studies; however, a subset of animals for EM should be considered and included as part of the experimental design in future studies.

[Editors' note: further revisions were suggested prior to acceptance, as described below.]

The manuscript has been improved but there are some remaining issues that need to be addressed, as outlined below:Revisions Required:1) When the term "non-linear change" is first introduced, it would be helpful to explain that it means non-linear with respect to time from the start of the learning paradigm.

We have now included an explanation of the term "non-linear change" both when it is first introduced in the Introduction and once more in the Introduction for clarity.

2) The term "length density" could also be defined on first use. Presumably it means length of myelin sheath per unit tissue volume. It could also be explained that this does not distinguish between elongation of existing myelin sheaths and addition of new myelin on previously unmyelinated regions of axons, either by newly generated or pre-existing oligodendrocytes.

We have also included an explanation of the term "length density" when it is first introduced.

3) How is myelin sheath thickness determined from the data illustrated in Figure 6? If it is from direct (or automated) measurement of sheath diameters in high-mag confocal images of MBP-labelled sections, then some representative high-mag images should be shown.

Myelin sheath thickness, or diameter of myelinated axons as illustrated in Figure 6 is from an automated measurement of sheath diameters from skeletonized MBP immunoreactivity in high-magnification confocal images. The script used in Amira automatically calculates the diameter in mm based upon the magnification factor and dimensions of the input data. We have also included in the text additional information (Material and Methods section) regarding the automated measurement of sheath diameter in the revised manuscript. An additional Supplementary Figure for Figure 6 is also added and accompanied with a figure legend.

4) In the Discussion (e.g., line 426) "formation of new cortical circuits" could perhaps be better described as "increased or de novo myelination of axons in cortical circuits" since the neuronal circuits/synaptic connections are presumably already in place at the start of training? Similarly (line 444) "...existing...connectivity that is enhanced and re-wired..." might be better: "...existing… circuits that are strengthened or modified through adaptive myelination."

We agree with the Editors and the Reviewers and have changed both instances in the Discussion.

5) The Discussion of adaptive myelination is framed in terms of wiring or re-wiring particular circuits that drive specific aspects of motor behaviour, rather like the wiring diagram of an electronic appliance (e.g., lines 471-473). However, another way of thinking about the problem is in terms of the potential effects of myelination on synchronization of bulk rhythmic activity within or between brain regions. Maybe this alternative view could also be mentioned? Arguably, it might be more compatible with the sort of macrostructural alterations that are observable using MRI?

We agree with the Editors and the Reviewers and have adapted the Discussion to include this alternative view and to reference Noori R *et al.*, 2020 (Activity-dependent myelination: A glial mechanism of oscillatory self-organization in large-scale brain networks, PNAS, 10.1073/pnas.1916646117).